# Personalized Algorithmic Recourse with Preference Elicitation

**Giovanni De Toni**  *giovanni.detoni@unitn.it*
*Augmented Intelligence Center, Fondazione Bruno Kessler, Italy*
*DISI, University of Trento, Italy*

**Paolo Viappiani**  *paolo.viappiani@lamsade.dauphine.fr*
*LAMSADE, CNRS, Université Paris-Dauphine, PSL, France*

**Stefano Teso**  *stefano.teso@unitn.it*
*CIMeC & DISI, University of Trento, Italy*

**Bruno Lepri**  *lepri@fbk.eu*
*Augmented Intelligence Center, Fondazione Bruno Kessler, Italy*

**Andrea Passerini**  *andrea.passerini@unitn.it*
*DISI, University of Trento, Italy*

**Reviewed on OpenReview:** *https://openreview.net/forum?id=8sg2I9zXgO*

## Abstract

*Algorithmic Recourse* (AR) is the problem of computing a *sequence of actions* that – once performed by a user – overturns an undesirable machine decision. It is paramount that the sequence of actions does not require too much effort for users to implement. Yet, most approaches to AR assume that actions cost the same for all users, and thus may recommend unfairly expensive recourse plans to certain users. Prompted by this observation, we introduce PEAR, the first human-in-the-loop approach capable of providing *personalized* algorithmic recourse tailored to the needs of any end-user. PEAR builds on insights from Bayesian Preference Elicitation to iteratively refine an estimate of the costs of actions by asking *choice set* queries to the target user. The queries themselves are computed by maximizing the *Expected Utility of Selection*, a principled measure of information gain accounting for uncertainty on both the cost estimate and the user's responses. PEAR integrates elicitation into a Reinforcement Learning agent coupled with Monte Carlo Tree Search to quickly identify promising recourse plans. Our empirical evaluation on real-world datasets highlights how PEAR produces high-quality personalized recourse in only a handful of iterations.

## 1 Introduction

Automated decision support systems are increasingly employed in high-risk decision tasks with the aim of empowering human decision-makers and improving the quality of their decisions. Example applications include bail requests (Dressel & Farid, 2018), loan approvals (Sheikh et al., 2020), job applications (Liem et al., 2018), and prescription of medications and treatments (Yoo et al., 2019). Despite their promise, often these systems are opaque – meaning that users, and even engineers, have trouble understanding and controlling their decision process – and provide no means for overturning unwanted outcomes, such as denied loan requests. One way of addressing these issues is through the lens of *Algorithmic Recourse* (AR) (Venkatasubramanian & Alfano, 2020; Karimi et al., 2021). In AR, given an undesirable machine-generated decision, the goal is to identify a sequence of actions – or *interventions* for short – that once implemented by the user overturns said decision, for instance, changing job or obtaining a master's degree. Motivated by this, a number of

approaches have been recently proposed for computing AR (De Toni et al., 2023; Naumann & Ntoutsi, 2021; Karimi et al., 2022; Ramakrishnan et al., 2020; Yonadav & Moses, 2019; Ustun et al., 2019; Karimi et al., 2020; Tsirtsis & Rodriguez, 2020; Russell, 2019; Mothilal et al., 2020; Wang et al., 2023; Dandl et al., 2020).

It is critical that the suggested recourse plans are not too difficult or expensive to carry out. This entails that recourse should be *personalized*, because different users in the same situation may *need* substantially different recourse plans. To see this, consider a user who is denied a loan. Based on a profile made by the financial institution, an AR algorithm might suggest the user reduce their monthly expenses. However, unlike the "average" customer, our user is incurring high medical expenses because they recently contracted an invalidating illness. Thus, the AR suggestion is highly inappropriate. Clearly, it is impossible to infer such a constraint from their profile alone. Most approaches, however, completely neglect the user's own preferences. The few that do require feedback that is difficult to obtain in practice, *e.g.*, preferences over a large pool of alternatives (Russell, 2019; Mothilal et al., 2020; Wang et al., 2023; Dandl et al., 2020) or upfront quantification of action costs (Yadav et al., 2021; Karimi et al., 2022; Rawal & Lakkaraju, 2020; Wang et al., 2023; Mahajan et al., 2019).

We argue that algorithmic recourse should make users first-class citizens in the recourse generation process rather than viewing them as passive observers. To this end, we introduce `PEAR` (Preference Elicitation for Algorithmic Recourse), the first human-in-the-loop approach for generating *personalized* recourse tailored for a target end-user. Our algorithm integrates AR and ideas from interactive Preference Elicitation (PE) (Chajewska et al., 2000; Boutilier, 2002; Braziunas & Boutilier, 2007; Guo & Sanner, 2010; Viappiani & Boutilier, 2010; Dragone et al., 2018) in a fully Bayesian setup. `PEAR` goes beyond existing approaches in that the costs of actions are estimated from user feedback and prior information. In each iteration, `PEAR` identifies a *small* selection of alternative interventions – a *choice set* – that optimizes a sound measure of information gain (the *Expected Utility of Selection* (`EUS`) (Viappiani & Boutilier, 2010; 2020)) and then asks the user to pick their preferred option. Using this feedback, `PEAR` quickly improves its estimate of the user's preferences and generates interventions that get progressively closer to the user's ideal. Furthermore, `PEAR` takes interactions and dependencies between actions costs into account when computing suggested recourse, whenever this information is available. See Fig. 1 for an overview.

**Contributions**: Summarizing, we:

- Introduce the problem of *personalized algorithmic recourse*, and show that existing approaches are insufficient to solve it.

- Develop `PEAR`, the first human-in-the-loop, Bayesian approach for computing *personalized* interventions that is robust to noise in user feedback and minimizes user effort.

- Evaluate `PEAR` on synthetic and real-world datasets and show that it can generate substantially – up to 50% – cheaper interventions than user-agnostic competitors after only a handful of queries.

## 2 Problem Statement

The user *state* $\mathbf{s} = (s_1, \ldots, s_d) \in \mathcal{S} \subseteq \mathbb{R}^d$ is a vector of $d$ categorical and real-valued features encoding, *e.g.*, instruction level and income. An *action* $a \in \mathcal{A}$ is a map that takes a state $\mathbf{s}$ and changes a *single* feature, yielding a new state $\mathbf{s}' = a(\mathbf{s})$ and expresses a recommendation of the form "*Increase your income by \$100.*" Given a (black-box) binary classifier[1] $h : \mathcal{S} \to \{0, 1\}$ and a user state $\mathbf{s}$ leading to an undesirable decision $y = h(\mathbf{s})$, AR computes an *intervention* – *i.e.*, an ordered set of actions $I = \{a^{(1)}, \ldots, a^{(|I|)}\}$ – that can be applied to $\mathbf{s}$ to obtain a counterfactual state $\mathbf{s}'$ associated to a more desirable outcome $h(\mathbf{s}') \neq y$, *all while minimizing user effort*. We formalize the user effort required to perform an action via a *cost function* $C : \mathcal{A} \times \mathcal{S} \to \mathbb{R}^+$. The cost of an intervention is the sum of the costs of all actions it contains, that is $C(I) = \sum_{i=0}^{|I|} C(a^{(i)}, \mathbf{s}^{(i)})$. Here, $\mathbf{s}^{(i)}$ is the state obtained by applying action $a^{(i-1)}$ to state $\mathbf{s}^{(i-1)}$, and $\mathbf{s}^{(0)} = \mathbf{s}$ is the initial state. We denote with $I(\mathbf{s}^{(0)})$ the operation of applying each action $a \in I$ sequentially.

---

[1]It is straightforward to adapt our approach to deal with multiclass classification problems.

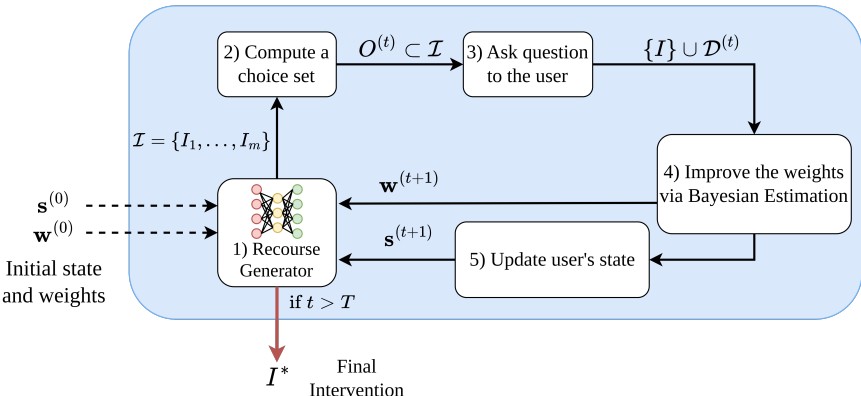

Figure 1: Overview of `PEAR`. (1) Given the initial state $\mathbf{s}^{(0)}$ of the user and weights $\mathbf{w}^{(0)}$, `PEAR` computes a pool of candidate interventions achieving recourse. (2) A choice set $O^{(t)}$ is selected from the pool and presented to the user. (3) The user picks their preferred intervention from the set. (4) An improved estimate of the weights $\mathbf{w}^{(t+1)}$ is computed using this feedback, and (5) the user's state $\mathbf{s}^{(t+1)}$ is updated. After $T$ rounds, the estimated weights are used to compute a final intervention $I^*$.

Approaches to AR assume the user effort is proportional to the number of actions that need to be carried out, and minimize it by searching for *short* interventions $I$. However, this is unrealistic and impractical. For instance, changing job into a highly skilled one may not be realistic without obtaining a Master's degree first. In reality, the user effort also depends on *users' preferences* $\mathbf{w} \in \mathcal{W} \subseteq \mathbb{R}^d$ which we represent as $d$-dimensional vectors. For example, for some people, it might be easier to improve their education than get new employment. Thus, we *parameterize* the cost function using the user preferences as $C(I \mid \mathbf{w})$, or $C(a, \mathbf{s} \mid \mathbf{w})$ if we refer to single actions. In the following, we distinguish between the (typically unknown) user preferences $\mathbf{w}^{GT}$ and the preferences $\mathbf{w}$ used by the recourse algorithm, which might differ from the former. Motivated by this, we introduce a new problem setting, denoted *personalized algorithmic recourse*:

**Definition 1 (Personalized Algorithmic Recourse)** *Given a black-box binary classifier $h$ and a user state $\mathbf{s}$, acquire a cost function $C(I \mid \mathbf{w})$ for interventions such that the intervention $I^*$ obtained by solving the following optimization problem:*

$$I^* \in \operatorname{argmin}_I \ C(I \mid \mathbf{w}) \quad \text{s.t.} \quad h(I(\mathbf{s}^{(0)})) \neq h(\mathbf{s}^{(0)}) \tag{1}$$

*has **minimal regret** for the target user, defined as:*

$$Reg(I^*, I^{GT}) = C(I^* \mid \mathbf{w}^{GT}) - C(I^{GT} \mid \mathbf{w}^{GT}) \tag{2}$$

*where $\mathbf{w}^{GT} \in \mathcal{W}$ encodes the ground-truth but unobservable preferences of the user and $I^{GT}$ is the "ideal" intervention that would be obtained by solving Eq. (1) using $\mathbf{w}^{GT}$.*

The similarity of Eq. (1) to existing formulations of AR can be misleading, as here the key challenge is that of obtaining weights $\mathbf{w}$ that reflect the user's own preferences. We discuss how `PEAR` does so in Section 4.

## 3 Related work

*Counterfactual explanations* (CEs) are a class of local, human-understandable explanations (Wachter et al., 2017; Byrne, 2019) that convey information about changes to input variables that overturn a machine decision (Guidotti et al., 2018; Stepin et al., 2021). AR aims to identify *actionable* CEs that attain recourse for the user (Verma et al., 2020; Karimi et al., 2022). Existing approaches to AR solve Eq. (1) via diverse optimization methods (Wachter et al., 2017; Ramakrishnan et al., 2020; Poyiadzi et al., 2020; Naumann & Ntoutsi, 2021; Russell, 2019; Mothilal et al., 2020; Wang et al., 2023; Dandl et al., 2020) or by learning a general policy

(Yonadav & Moses, 2019; De Toni et al., 2023; Verma et al., 2022). Most methods simply return a set of actions, disregarding their order. However, recent research showed that ignoring the causal relationship between features prevents reaching optimal recourse (Karimi et al., 2021). Some methods thus optimize for recourse plans, *i.e.*, *sequences* of actions attaining recourse, following a purely causal or causal-inspired setup (Ustun et al., 2019; Karimi et al., 2021; 2020; De Toni et al., 2023; Naumann & Ntoutsi, 2021; Mahajan et al., 2019). PEAR fully supports this paradigm and considers the interplay between features when finding recourse, whenever this information is available.

Most AR approaches assume that the cost function is fully specified beforehand (Wachter et al., 2017; Ramakrishnan et al., 2020; Rawal & Lakkaraju, 2020; Wang et al., 2023; Mahajan et al., 2019), ignoring the problem of modelling user preferences altogether. The few that explicitly deal with user preferences do so in a naïve manner. Some of them (Russell, 2019; Mothilal et al., 2020; Dandl et al., 2020; Yadav et al., 2021) ask users to pick their preferred option from a *large* pool of user-agnostic recourse plans, that is not guaranteed to contain a low-cost option for the user. This is also impractical, as users can only properly evaluate a limited number of alternatives at a time (Simon, 1955; March, 1978). Others require users to *quantify* the cost of each possible action upfront (Karimi et al., 2022; Rawal & Lakkaraju, 2020; Wang et al., 2023; Mahajan et al., 2019), or via numerical constraints (Poyiadzi et al., 2020; Wang et al., 2023), yet end-users can rarely articulate their preferences in a quantitative manner (Keeney & Raiffa, 1993). PEAR sidesteps this issue by learning preferences from ranking data, *i.e.*, relative judgments of the form "I prefer option $A$ to option $B$" (Furnkranz & Hullermeier, 2010).

AR is specifically concerned with high-stakes scenarios, such as loan requests, that users face at most a handful of times in their lifetime. In these settings, it is impossible to estimate user preferences from historical data. In recommender systems, this issue has been solved through *preference elicitation*, whereby the user's preferences are estimated through a human-friendly interaction protocol (Boutilier, 2002). In order to converge to high-quality options with minimal user effort, PE algorithms select queries (*i.e.*, questions to the user) that maximize information gain and that are easy to answer. A popular option is *choice queries*, in which the user has to select a preferred item from a small set of alternatives. PEAR builds on Bayesian PE methods, as they account for imprecision in users' answers (Chajewska et al., 2000; Guo & Sanner, 2010; Viappiani & Boutilier, 2010) in a principled way by measuring the information gain of choice sets in terms of Expected Utility of Selection (Viappiani & Boutilier, 2020). Similarly to PEAR, Rawal & Lakkaraju (2020) estimate cost weights from preference feedback. However, they collect feedback from *domain experts* in a non-interactive fashion and learn *population-level* preferences that are not personalized, thus failing to minimize Eq. (2). Population-level estimates can lead to recourse that is largely suboptimal for specific individuals, as will be shown in our experimental evaluation.

## 4  Personalized Algorithmic Recourse with PEAR

Before describing PEAR, we discuss how we model the user's preferences and how these impact the costs of actions and interventions. Equally importantly, actions influence each other's costs – *e.g.*, obtaining an additional degree can dramatically lower the cost of landing a better-paying job – and we need to account for this if we wish to minimize the cost of recourse. In order to account for interactions between action costs, we model $C$ using a set of linear equations, parameterized by weights $\mathbf{w} \in \mathcal{W}$, by taking inspiration from *generalised additive independence* models (Pigozzi et al., 2016) from decision theory and *structural causal models* (Pearl, 2009). We call such formalization *cost correlation structure* (CCS). A CCS corresponds to a directed acyclic graph (DAG) where each node is associated with the cost $w_k$ of changing a single feature $s_k$ and each edge represents a (causal) relationship between two costs, parameterized by $w_{jk} \in \mathbb{R}$. Then, following the equation induced by the DAG, we define the cost of applying an action $a_k$ to a state $\mathbf{s}$ to change the $k$-th feature from $s_k$ to $s'_k$ as a linear combination:

$$C(a_k, \mathbf{s} \mid \mathbf{w}) = w_k |s'_k - s_k| + \sum_{j \in Pa_k} w_{jk} s_j \tag{3}$$

Here, $Pa_k$ are the *parents* of the $k$-th node in the graph. For example, we can use Eq. (3) to describe the cost of the action "*increase your salary*". Such cost depends linearly on the raise asked ($|s'_k - s_k|$). However, it also depends on some causally related variables ($\sum_{j \in Pa_k} w_{jk} s_j$), such as the user's current job. We can

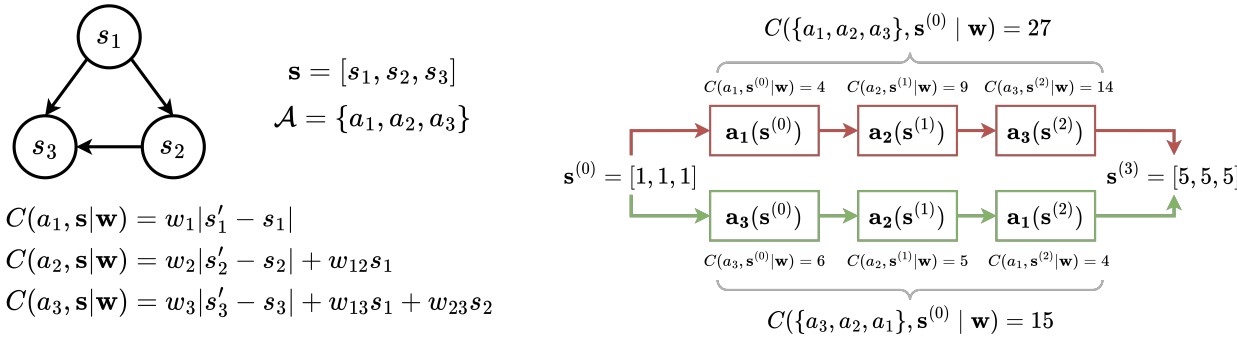

(a) Cost Correlation Structure (CCS)          (b) The cost of an intervention $I$

Figure 2: **The cost model** (Left) A *cost correlation structure* (CCS) for cost modelling. (Right) Given $\mathbf{s}^{(0)} = [1, 1, 1]$ and unit $\mathbf{w}$, let us imagine we want to reach $\mathbf{s}^{(3)} = [5, 5, 5]$ by following the presented CCS and the intervention $I = \{a_1, a_2, a_3\}$, where $a_i$ assign $s_i \leftarrow 5$ for all $i \in \{1, 2, 3\}$. Clearly, we incur different costs by applying permuted versions of $I$. The green path indicates the lower-cost intervention.

imagine that it might be easier to ask for a raise for a company's manager rather than a simple employee. A formal example of a cost function and of its corresponding DAG is shown in Fig. 2a. Again, the cost of an intervention is the sum of the costs of all actions it contains applied sequentially, that is:

$$C(I \mid \mathbf{w}) = \sum_{i=0}^{|I|} C(a^{(i)}, \mathbf{s}^{(i)} \mid \mathbf{w}) \tag{4}$$

Eq. (3) does not define a simple linear model, but it accounts for a richer set of interactions. For example, given two actions $a_1$ and $a_2$, their cost is not additive, meaning $C(a_1, \mathbf{s} \mid \mathbf{w}) + C(a_2, \mathbf{s} \mid \mathbf{w}) \neq C(\{a_1, a_2\}, \mathbf{s} \mid \mathbf{w})$. We can show it with a simple example.

**Example of non-additivity of Eq. (3).** Consider a simple CCS with two nodes $s_1 \rightarrow s_2$, with $\mathbf{s} = (s_1, s_2)$ and $\mathbf{w} = (w_1, w_2, w_{12})$, and an instance $\mathbf{s} = (1, 1)$ and $\mathbf{w} = (1, 0.5, 1)$. We define two actions, $a_1$ and $a_2$, which simply set the corresponding features $s_i$ to 2. Using Eq. (3), we can compute the cost of applying the single action as

$$C(a_1, \mathbf{s} \mid \mathbf{w}) = 1|2 - 1| = 1 \qquad C(a_2, \mathbf{s} \mid \mathbf{w}) = 0.5|2 - 1| + 1 = 1.5 \tag{5}$$

By Eq. (4), the cost of the intervention $\{a_1, a_2\}$ is $C(\{a_1, a_2\}, \mathbf{s} \mid \mathbf{w}) = 3.5 \neq 1.5 + 1$, hence the costs are *not* additively independent. Moreover, Eq. (4) considers the *order* in which actions are applied, *i.e.*, $\exists \, \mathbf{w}$ such that $C(\{a_1, a_2\}, \mathbf{s} \mid \mathbf{w}) \neq C(\{a_2, a_1\}, \mathbf{s} \mid \mathbf{w})$, as shown in Fig. 2b.

Note that if dependencies between actions costs are not available, Eq. (3) boils down to a simple linear function, namely $C(a_k, \mathbf{s} \mid \mathbf{w}) = w_k|s'_k - s_k|$. This case matches the typical setup of algorithmic recourse methods, which assume the weights are independent from one another. Eq. (3) is however more flexible in that, *if* dependencies between costs *are* in fact available, they are automatically leveraged by PEAR.

We formalized the cost of an action (Eq. (3)) by following a standard assumption made in decision theory (Keeney & Raiffa, 1993) called "*generalised additive independence*" (GAI) (Pigozzi et al., 2016) which is shared by many preference elicitation algorithms. It defines utility functions that measure both the contribution of a single feature and the contribution of a subset of features. Following causality theory (Pearl, 2009), each subset consists of features causally related to the one we are acting on. In support of this strategy, previous research highlighted how it is impossible to offer optimal recourse recommendations without considering relationships between features (Karimi et al., 2020).

In Karimi et al. (2020), the authors refer to knowing the impact of an intervention on causally related features (e.g., will getting a new degree increase my salary?). However, it is known that such causal knowledge about the world is hard to obtain unless we make specific restricting assumptions (Pearl, 2009). In our work, rather than considering how features causally change after an intervention, we focus on learning how the "cost"

of acting on those features changes after an intervention (e.g., will getting a new degree make it easier to increase my salary?). In our case, the cost of modifying a feature is something we can easily learn by asking the user.

Ultimately, with our setup, we make learning the cost function feasible, and we can also represent complex non-additive behaviour which arises from the sequential nature of recourse. Our formalization in Eq. (3) and Eq. (4) goes along the lines of previous works (Naumann & Ntoutsi, 2021; De Toni et al., 2023), that address the problem from the same perspective, but assume to know these costs a-priori.

**On Cost Correlation Structure and Causality.** The *cost correlation structure* shares some similarities with Structural Causal Models (SCMs) (Pearl, 2009), since they both use a directed acyclic graph, also called *causal graph*, to represent causal relationships between features $\mathbf{X}$. In the context of AR, SCMs model relationships between *features* rather than between *costs*. For example, Karimi et al. (2020) frames the recourse problem as finding a counterfactual $\mathbf{s}^{\mathbb{CF}}$ optimizing Eq. (1) given an approximated SCM describing the joint distribution $P(\mathbf{X})$, while the cost function is a simple $p$-norm, such as $C(\mathbf{s}^{\mathbb{CF}}, \mathbf{s}) = \|\mathbf{s}^{\mathbb{CF}} - \mathbf{s}\|_2$. In our case, CCSs do not allow for *do*-actions or to compute counterfactuals via abduction, since they do not model $P(\mathbf{X})$. Similarly to Naumann & Ntoutsi (2021), we use the causal DAG to represent consequence-aware cost equations as shown in Eq. (3), without taking any explicit causal angle. Indeed, following the non-causal AR literature, we assume actions $a \in \mathcal{A}$ to influence only the target feature $s_k$, without any causal effect on any of its children $s_j$ in the DAG. However, actions over $s_k$ will affect the *cost* of subsequent actions changing the user features.

## 4.1 The `PEAR` Algorithm

In order to account for uncertainty over the user's weights $\mathbf{w}$, `PEAR` explicitly models a distribution $P(\mathbf{w})$ over them and progressively refines it by interacting with a target user. A high-level overview of `PEAR` is given in Fig. 1 and the pseudo-code is listed in Algorithm 1.

In each iteration $t = 1, \ldots, T$, where $T$ is the iteration budget, `PEAR` computes a *choice set* $O^{(t)} \in \mathcal{I}^k$ containing $k$ candidate interventions achieving recourse (for a small $k$, *e.g.*, 2 to 4) and asks the user to indicate their most preferred option in the set. Importantly, $O^{(t)}$ is chosen so as to maximize the (expected) information gained from the user, and in a way that is robust to noise in their feedback. We detail the exact procedure used by `PEAR` in Section 4.3. These user choices are stored in an initially empty dataset $\mathcal{D}^{(t)}$. In each step, `PEAR` integrates the user's feedback by inferring a posterior over the weights $P(\mathbf{w} \mid \mathcal{D}^{(t)}) \propto P(\mathcal{D}^{(t)} \mid \mathbf{w})P(\mathbf{w})$ using Bayesian inference, and updates the user state by applying the first action $\hat{I}_1$ of the chosen intervention. We apply a single action so as to elicit user preferences in all intermediate states. If a state achieving recourse is reached, the user state is reinitialized. After $T$ rounds,[2] `PEAR` computes a low-cost personalized intervention by applying the intervention generation procedure described in Section 4.2, biased according to the latest posterior $p(\mathbf{w} \mid \mathcal{D}^{(t)})$.

`PEAR` makes no assumption on the form of the prior $P(\mathbf{w})$, meaning that the prior can be adjusted based on the application. In order to model both variances across the preferences of individuals and for sub-groups in the population, in this work we model them as a mixture of Gaussians with $M$ components $\mathcal{N}_i(\mu_i, \mathbf{\Sigma}_i)$, for $i = 1, \ldots, M$. This choice works well in our experiments, see Section 5. Note that, analogously to Rawal & Lakkaraju (2020), it is also possible to fit the prior on population-level preference data or domain expert input, whenever this is available. In our experiments, we do this for *all* competitors.

## 4.2 Generating Personalized Interventions with `W-FARE`

`PEAR` generates personalized interventions by leveraging a novel, *user-aware* extension of `FARE` (De Toni et al., 2023), a state-of-the-art algorithm for generating short – but *user-agnostic* – interventions, which we briefly outline next. In `FARE`, each action $a \in \mathcal{A}$ is implemented as a tuple $(f, x)$, where $f$ is a function changing *one* feature and $x$ is the value that feature takes, *e.g.*, (*change_income*, \$1000). Given an initial state $\mathbf{s}$, `FARE` uses reinforcement learning to learn two probabilistic policies $\pi_f(\mathbf{s})$ and $\pi_x(\mathbf{s})$, which are used as priors to guide a Monte Carlo Tree Search procedure that incrementally builds an intervention $I$ by selecting actions

---

[2]In practice, the loop can be terminated as soon as the user is satisfied with one of the interventions in $O^{(t)}$.

---

**Algorithm 1** The PEAR algorithm: $h : \mathcal{S} \to \{0, 1\}$ is a classifier, $\mathbf{s}^{(0)} \in \mathcal{S}$ the initial state, $\mathcal{A}$ the available actions, $p(\mathbf{w})$ the prior, $T \geq 1$ the query budget, $k \geq 2$ is the size of choice sets.

---

1: **procedure** PEAR($h, \mathbf{s}^{(0)}, \mathcal{A}, T, k$)
2:     Initialize $t \leftarrow 0$, $\mathcal{D}^{(0)} \leftarrow \varnothing$
3:     **for** $t = 1, \ldots, T$ **do**
4:         $O^{(t)} \leftarrow$ SUBMOD-CHOICE($h, \mathbf{s}^{(t-1)}, \mathcal{A}, k, \mathcal{D}^{(t-1)}$)                  ▷ Algorithm 2
5:         Ask the user to pick the best intervention $\hat{I} \in O^{(t)}$
6:         $\mathcal{D}^{(t)} \leftarrow \mathcal{D}^{(t-1)} \cup \{\hat{I}\}$
7:         Update weight estimate $p(\mathbf{w} \mid \mathcal{D}^{(t)})$
8:         $\mathbf{s}^{(t)} \leftarrow \hat{I}_1(\mathbf{s}^{(t-1)})$
9:         **if** $h(\mathbf{s}^{(t)}) \neq h(\mathbf{s}^{(0)})$ **then**
10:             $\mathbf{s}^{(t)} \leftarrow \mathbf{s}^{(0)}$
11:     $I^* =$ W-FARE($h, \mathbf{s}^{(0)}, \mathbf{w}^*$) with $\mathbf{w}^* = \mathbb{E}_{P(\mathbf{w}|\mathcal{D}^{(T)})}[\mathbf{w}]$             ▷ Section 4.2
12:     **return** $I^*$

---

$a^{(i)} \in \mathcal{A}$. In order to ensure interventions are *actionable*, actions $a$ are only chosen if they satisfy given preconditions. The reward used by FARE is $r(I) = \rho^{|I|} \cdot \mathbb{1}\{h(I(\mathbf{s}^{(0)})) \neq h(\mathbf{s}^{(0)})\}$, where $\rho > 0$ is a discount factor and the indicator evaluates to 1 if $I$ attains recourse and to 0 otherwise. FARE is highly scalable and very effective at identifying counterfactual interventions even under minimal training budget (De Toni et al., 2023).

FARE is user-agnostic, while PEAR needs to generate *personalized* interventions. We fill this gap by introducing W-FARE, a novel extension of FARE that integrates the user's costs into the reward while inheriting all benefits of the latter. Recall that PEAR maintains a posterior over the weights. The *expected cost of an action* can thus be obtained by marginalizing over the posterior:

$$\mathbb{E}[C(a, \mathbf{s}) \mid \mathcal{D}^{(t)}] = \int_{\mathbf{w}} C(a, \mathbf{s} \mid \mathbf{w}) P(\mathbf{w} \mid \mathcal{D}^{(t)}) \, d\mathbf{w} \tag{6}$$

Analogously, the cost of an intervention $I$ is replaced by the expectation:

$$\mathbb{E}[C(I) \mid \mathcal{D}^{(t)}] = \sum_{i=0}^{|I|} \mathbb{E}[C(a^{(i)}, \mathbf{s}^{(i)}) \mid \mathcal{D}^{(t)}] \tag{7}$$

The W-FARE reward function is then given by $r(I \mid \mathbf{w}) \propto \rho^{\mathbb{E}[C(I|\mathcal{D}^{(t)})]} \cdot \mathbb{1}\{h(I(\mathbf{s}^{(0)})) \neq h(\mathbf{s}^{(0)})\}$ (see Appendix B for a more in-depth description of W-FARE). This explicitly drives RL to learn policies that optimize user-specific action costs and that, therefore, help MCTS to more quickly converge to *personalized* interventions. We show empirically that PEAR is substantially more effective than FARE at computing personalized interventions in Section 5.

### 4.3 Computing Informative Choice Sets

Given the current posterior $P(\mathbf{w} \mid \mathcal{D}^{(t)})$, PEAR computes a *choice set* containing $k$ interventions $I$ that maximizes information gain (Chajewska et al., 2000). We measure the latter using the *Expected Utility of Selection* (EUS) (Viappiani & Boutilier, 2020), a measure of the goodness of a set defined as the expectation, under the uncertainty over $\mathbf{w}$, of the utility of its most preferred element. EUS is closely related to the Expected Value of Information (EVOI), and frequently used in Bayesian PE (Price & Messinger, 2005; Viappiani & Boutilier, 2010; Akrour et al., 2012; Viappiani & Boutilier, 2020). The EUS builds on the notion of *expected utility of an intervention* $I$, which is defined as:

$$\mathsf{EU}(I \mid \mathcal{D}^{(t)}) = \mathbb{E}[-C(I) \mid \mathcal{D}^{(t)}] = - \int_{\mathbf{w}} C(I \mid \mathbf{w}) P(\mathbf{w} \mid \mathcal{D}^{(t)}) \, d\mathbf{w} \tag{8}$$

The EUS of a choice set $O$ can then be defined as:

$$\begin{aligned} \mathsf{EUS}_R(O \mid \mathcal{D}^{(t)}) &= \sum_{I \in O} P_R(O \rightsquigarrow I) \mathsf{EU}(I \mid \mathcal{D}^{(t)}) \\ &= - \int_{\mathbf{w}} \left[ \sum_{I \in O} P_R(O \rightsquigarrow I \mid \mathbf{w}) C(I \mid \mathbf{w}) \right] P(\mathbf{w} \mid \mathcal{D}^{(t)}) \, d\mathbf{w} \end{aligned} \tag{9}$$

---

**Algorithm 2** Greedy procedure to efficiently compute a choice set $O$: $\mathbf{s}^{(t)} \in \mathcal{S}$ the current state, $\mathcal{A}$ the available actions, $k \geq 2$ is the size of choice sets, $\mathcal{D}^{(t)}$ the user choices so far.

1: **procedure** SUBMOD-CHOICE($\mathbf{s}^{(t)}, k, \mathcal{A}, \mathcal{D}^{(t)}$)
2:     $O \leftarrow \varnothing$
3:     $\bar{\mathbf{w}} \leftarrow \mathbb{E}_{p(\mathbf{w}|\mathcal{D}^{(t)})}[\mathbf{w}]$
4:     **while** $|O| < k$ **do**
5:         Generate the candidate interventions $\mathcal{I}$ with W-FARE using $\mathcal{A}$ and $\bar{\mathbf{w}}$
6:         $\hat{I} \leftarrow \text{argmax}_{\hat{I}} \ \mathsf{EUS}_{NL}(O \cup \hat{I} \mid \mathcal{D}^{(t)}) - \mathsf{EUS}_{NL}(O \mid \mathcal{D}^{(t)})$
7:         $O \leftarrow O \cup \{\hat{I}\}$
8:     **return** $O$

---

Here, $P_R(O \rightsquigarrow I \mid \mathbf{w})$ is the probability that a user with weights $\mathbf{w}$ picks $I$ from $O$, under a specific choice of *response model* $R$ modelling noise in user choices. Intuitively, we expect users to prefer the cheapest interventions $I \in O$. Moreover, we also expect that interventions in $O$ with similar costs have a similar probability of being chosen. Motivated by this, and following common practice in choice modelling (Luce, 2012), in PEAR we implement a *logistic response model* ($L$), defined as:

$$P_L(O \rightsquigarrow I \mid \mathbf{w}) = \frac{\exp(-\lambda C(I \mid \mathbf{w}))}{\sum_{I \in O} \exp(-\lambda C(I \mid \mathbf{w}))} \tag{10}$$

Here, $\lambda \in \mathbb{R}$ is a temperature parameter. Finding a choice set $O$ maximizing the EUS is intractable in general – *NP-hard* (Nemhauser et al., 1978; Krause & Golovin, 2014), in fact – and computationally intensive in practice, and risks slowing down the interaction loop to the point of estranging users. We observe that, however, under some response models $R$, the EUS becomes *submodular* and *monotonic* (Viappiani & Boutilier, 2010). This is the case for the *noiseless* response model ($NL$), according to which the user always prefers the lowest-cost option, *i.e.*,[3]

$$P_{NL}(O \rightsquigarrow I \mid \mathbf{w}) = \prod_{I,I' \in O \,:\, I \neq I'} \mathbb{1}\{C(I \mid \mathbf{w}) < C(I' \mid \mathbf{w})\} \tag{11}$$

This means that, for *NL*, greedy optimization is sufficient to find a choice set $O$ that achieves high $\mathsf{EUS}_{NL}$ with approximation guarantees. Formally, it holds that for choice sets $O$ found via greedy optimization, $\mathsf{EUS}_{NL}(O \mid \mathcal{D}) \geq (1 - e^{-1})\mathsf{EUS}_{NL}(O^* \mid \mathcal{D})$, where $O^*$ is the truly optimal choice set (Viappiani & Boutilier, 2010; Nemhauser et al., 1978; Krause & Golovin, 2014). In PEAR, we leverage the fact that $\mathsf{EUS}_L - \mathsf{EUS}_{NL}$ is always smaller than a problem independent (tight) bound (Viappiani & Boutilier, 2010), meaning that instead of minimizing $\mathsf{EUS}_L$ directly, we can compute a high-quality choice set by greedily maximizing $\mathsf{EUS}_{NL}$. This immediately leads to a practical algorithm for the logistic response model $L$, listed in Algorithm 2.

### 4.4 Benefits and limitations

PEAR is designed to facilitate the application of AR to practical high-stakes tasks like loan approval. The main benefit of PEAR is that it provides *personalized* algorithmic recourse, which existing approaches are not capable of. Also, it follows a fully Bayesian setup for handling uncertainty over the estimated preferences and noise in user feedback. It also leverages ideas from preference elicitation – such as small *choice sets* and elicitation of relative preferences – to ensure the interaction is cognitively affordable. Finally, PEAR makes use of a novel cost function that automatically takes dependencies between action costs into consideration, whenever these are known or can be estimated, thus facilitating the identification of better recourse suggestions. At the same time, the cost function is linear, enabling PEAR to leverage efficient algorithms from preference elicitation tailored for eliciting such cost functions.

Several steps of the algorithm – namely, evaluating the EUS (Eq. (9)) and the expected cost of interventions (Eq. (7)), and updating the posterior $p(\mathbf{w} \mid \mathcal{D}^{(t)})$ – require marginalizing over the weights. Doing so involves evaluating a complex integral that cannot be solved analytically. We sidestep this issue by leveraging

---

[3]For the sake of presentation, we assume that there are no ties. Note that the EUS formula is invariant to the way ties are broken. In our implementation, ties are broken uniformly at random.

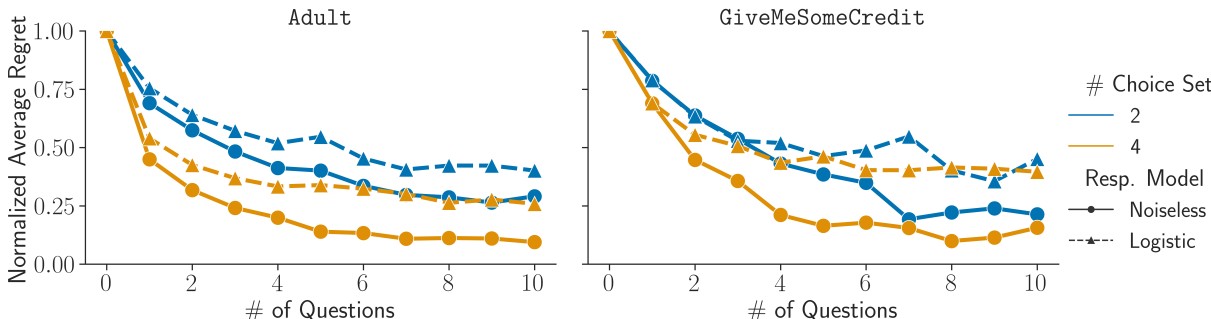

Figure 3: Normalized Average Regret for `PEAR` when varying the number of questions, the choice set size and the user response model on both datasets (sampled from `All` users).

an efficient Monte Carlo approximation, and specifically *ensemble split sampling* (Karamanis & Beutler, 2020; Karamanis et al., 2021) and then averaging over the samples with the highest likelihood. We find that, empirically, this procedure is efficient – so much that it can support interactive usage – and leads to competitive results in our experiments, cf. Section 5.

Lastly, as is typical in PE, our approach focuses on the "myopic" value of information by looking to maximize the expected posterior utility, thus reducing as much as possible the regret *one step ahead*. As we ask more questions, the regret will tend to zero. However, one main theoretical challenge currently faced in the PE domain is the development of optimal elicitation policies that consider a *sequential* version of the value of information, which may be more efficient but intractable in general (Boutilier, 2002).

## 5 Experiments

Our experimental evaluation is aimed at answering the following research questions:

**Q1** Does `PEAR` succeed in minimizing regret for an increasing amount of user feedback?

**Q2** Does `PEAR` outperform competitors in terms of validity and cost?

**Q3** Is `PEAR` robust to imprecise knowledge of the cost correlation structure of the user?

**Datasets and Classifiers.** We evaluated our approach on two real-world datasets taken from the relevant literature: `GiveMeSomeCredit` (Kaggle, 2011) and `Adult` (Dua & Graff, 2017). They are (unbalanced) binary classification problems for income prediction and loan assignment, respectively. The datasets have both categorical and numerical features. Some of these features are actionable (e.g., *occupation*, *education*), while others are immutable (e.g., *age*, *sex*, *native_country*). These datasets come without a causal graph and users' preferences over the features. Following previous work (Karimi et al., 2021; De Toni et al., 2023; Naumann & Ntoutsi, 2021), we manually defined and fixed the cost correlation structures for both. We randomly generated user-specific weights for each instance by sampling from a (dataset-specific) mixture of Gaussians with $M = 6$ components and uniform mixture weights. In Appendix A, we give a description of the feature space, action space and the directed acyclic graphs needed by the CCS for both datasets. We then split the data into training (70%), validation (10%) and test (20%) sets. For each dataset, we designed the black-box classifier $h$ as an MLP with two hidden layers. We trained it by cross-entropy minimization, selecting the hyperparameters which maximise the $F_1$ score on the validation set. In Appendix C we show additional experiments with shallow classifiers, such as a logistic model and a decision tree.

**"Easy" vs. "Hard" Users.** Intuitively, users close to the decision boundary of the black-box $h$ will require few actions to achieve recourse, while users to whom $h$ assigns a low score might need longer and more complex interventions. Understanding the preferences of these "hard" users is crucial since a wrong suggestion might

Table 1: Performance of all competitors averaged over 10 runs. A '-' indicates that the method did not find *any* successful intervention for *any* user. $\texttt{PEAR}_{NL}$ and $\texttt{PEAR}_L$ indicate $\texttt{PEAR}$ associated with the noiseless and logistic response model, respectively. The best results are boldfaced.

| Users | Method | Adult | | | GiveMeSomeCredit | | |
|---|---|---|---|---|---|---|---|
| | | *Validity* | *Cost* | *Length* | *Validity* | *Cost* | *Length* |
| All | FARE | $0.90 \pm 0.27$ | $285.63 \pm 195.68$ | $3.45 \pm 1.19$ | $0.86 \pm 0.23$ | $161.77 \pm 107.42$ | $3.27 \pm 1.13$ |
| | CSCF | $0.78 \pm 0.28$ | $154.28 \pm 125.34$ | $2.53 \pm 0.57$ | $0.57 \pm 0.42$ | $100.69 \pm 120.22$ | $2.51 \pm 1.12$ |
| | EFARE | $0.76 \pm 0.39$ | $306.11 \pm 199.02$ | $3.54 \pm 1.25$ | $0.67 \pm 0.38$ | $155.92 \pm 109.19$ | $3.18 \pm 1.18$ |
| | RL | $0.76 \pm 0.38$ | $283.31 \pm 167.86$ | $3.36 \pm 1.08$ | $0.12 \pm 0.32$ | $59.66 \pm 0.00$ | $2.00 \pm 0.00$ |
| | MCTS | $0.44 \pm 0.44$ | $445.65 \pm 201.28$ | $4.60 \pm 1.19$ | $0.70 \pm 0.42$ | $214.33 \pm 119.38$ | $4.09 \pm 1.31$ |
| | FACE | $0.15 \pm 0.27$ | $397.49 \pm 128.43$ | $3.76 \pm 0.64$ | $0.24 \pm 0.38$ | $327.18 \pm 78.85$ | $5.97 \pm 0.62$ |
| | $\texttt{PEAR}_{NL}$ (ours) | $\mathbf{1.00 \pm 0.03}$ | $\mathbf{142.23 \pm 61.75}$ | $\mathbf{2.84 \pm 0.59}$ | $\mathbf{0.89 \pm 0.00}$ | $\mathbf{96.04 \pm 31.96}$ | $\mathbf{2.79 \pm 0.42}$ |
| | $\texttt{PEAR}_L$ (ours) | $\mathbf{1.00 \pm 0.04}$ | $\mathbf{146.54 \pm 63.09}$ | $\mathbf{2.84 \pm 0.56}$ | $\mathbf{0.89 \pm 0.00}$ | $\mathbf{100.19 \pm 29.53}$ | $\mathbf{2.85 \pm 0.48}$ |
| Hard | FARE | $0.71 \pm 0.44$ | $438.97 \pm 188.50$ | $4.54 \pm 1.21$ | $0.47 \pm 0.37$ | $319.46 \pm 96.36$ | $4.97 \pm 0.70$ |
| | EFARE | $0.55 \pm 0.48$ | $454.05 \pm 202.76$ | $4.52 \pm 1.25$ | $0.22 \pm 0.36$ | $371.58 \pm 82.18$ | $5.31 \pm 0.71$ |
| | RL | $0.55 \pm 0.47$ | $433.64 \pm 152.67$ | $4.33 \pm 1.24$ | - | - | - |
| | CSCF | $0.25 \pm 0.36$ | $382.84 \pm 126.31$ | $3.70 \pm 0.56$ | $0.13 \pm 0.32$ | $190.81 \pm 119.99$ | $3.36 \pm 1.11$ |
| | MCTS | $0.21 \pm 0.40$ | $599.20 \pm 153.44$ | $5.53 \pm 0.72$ | $0.40 \pm 0.43$ | $353.75 \pm 99.26$ | $5.43 \pm 0.88$ |
| | FACE | $0.00 \pm 0.04$ | $448.72 \pm 0.00$ | $5.20 \pm 0.00$ | $0.20 \pm 0.36$ | $455.20 \pm 92.10$ | $7.09 \pm 0.53$ |
| | $\texttt{PEAR}_{NL}$ (ours) | $\mathbf{0.99 \pm 0.08}$ | $\mathbf{296.37 \pm 43.84}$ | $\mathbf{3.35 \pm 0.55}$ | $\mathbf{0.58 \pm 0.04}$ | $\mathbf{251.60 \pm 51.16}$ | $\mathbf{4.59 \pm 0.40}$ |
| | $\texttt{PEAR}_L$ (ours) | $\mathbf{0.99 \pm 0.09}$ | $\mathbf{301.13 \pm 52.61}$ | $\mathbf{3.34 \pm 0.58}$ | $\mathbf{0.58 \pm 0.02}$ | $\mathbf{262.23 \pm 45.36}$ | $\mathbf{4.64 \pm 0.36}$ |

substantially increase the overall cost for them. For each dataset, we thus built two separate testing sets. The first one, named $\texttt{All}$, is obtained by sampling 300 users $\mathbf{s}$ with an unfavourable classification ($h(\mathbf{s}) < 0.5$), regardless of the actual value of $h$. The second one, named $\texttt{Hard}$, is obtained by sampling 300 users with an unfavourable classification having a score in the lower quartile of the black-box score distribution.

**Competitors.** We compare $\texttt{PEAR}$ against several baselines: $\texttt{FARE}$ and its explainable version $\texttt{EFARE}$ (De Toni et al., 2023), $\texttt{CSCF}$ (Naumann & Ntoutsi, 2021), an evolutionary algorithm which, similarly to $\texttt{FARE}$, generates recourse options by considering consequence-aware cost functions and action sets $\mathcal{A}$, and $\texttt{FACE}$ (Poyiadzi et al., 2020), a well-know AR algorithm, which optimizes for population-based "feasible paths" to achieve recourse. We also consider two simpler baselines, a brute-force search ($\texttt{MCTS}$) and a vanilla reinforcement learning agent ($\texttt{RL}$), trained in a similar way as in (Hamrick et al., 2019; Verma et al., 2022). Note that all the competitors are model-agnostic and *not* interactive, since they assume the users' costs to be fixed. We provide more in-depth details on the baselines and their implementations in Appendix A.

**Experimental Protocol.** For $\texttt{PEAR}$, we vary the number of questions $T$ to the user from 0 to 10. For $T = 0$, we initialize the weights with the expected value of the prior, $\mathbb{E}_{P(\mathbf{w})}[\mathbf{w}]$, that represents a user-independent population-based prior. Moreover, we employ two user response models, the *noiseless* model (Eq. (11)), to check the effectiveness of our approach in the best-case scenario where the user can perfectly express their preferences, and the *logistic* model (Eq. (10)), to challenge our approach in a more realistic scenario. To provide a fair comparison, we equip the competitors with our cost function (Eq. (3)) and set their weights to the expected value of the prior.

**Q1: PEAR successfully minimizes the regret.** Fig. 3 shows the evaluation of the regret as a function of the number of queries to the user. Here the ground-truth intervention $I^{GT}$ (which is unknown) is approximated by running $\texttt{PEAR}$ with the correct user costs $\mathbf{w}^{GT}$, and the regret is normalized by rescaling the costs between $C(I^{\text{GT}} \mid \mathbf{w}^{GT})$ and $C(I^{(0)} \mid \mathbf{w}^{GT})$ where we generate $I^{(0)}$ using the expectation of the prior. We run $\texttt{PEAR}$ with two different choice set dimensions, $k = 2$ and $k = 4$, and for both noiseless and logistic response models. After a few questions, $\texttt{PEAR}$ reaches a low regret in all settings. Generally, a larger choice set produces a lower regret, irrespective of the response model, with the downside of increasing the cognitive burden for the user. We now briefly summarize the results when $T = 10$. For the $\texttt{Adult}$ dataset, the best regret is $\approx 0.09$ for the noiseless user and $k = 4$, while the worst regret is $\approx 0.40$ for the logistic response model and $k = 2$. For

Table 2: Evaluation of `PEAR` (with $q = 10$ and a logistic noise model) for an increasing amount of CCS graph corruption, averaged over 10 runs. "None" indicates that the correct causal graph is being used.

| Users | Corruption | Adult | | | GiveMeSomeCredit | | |
|-------|-----------|-------------------|--------------------|----------------|-------------------|---------------------|----------------|
| | | *Validity* | *Cost* | *Length* | *Validity* | *Cost* | *Length* |
| All | **None** | $1.00 \pm 0.04$ | $146.54 \pm 63.09$ | $2.84 \pm 0.56$ | $0.89 \pm 0.00$ | $100.19 \pm 29.53$ | $2.85 \pm 0.48$ |
| | **0.15** | $1.00 \pm 0.00$ | $165.11 \pm 46.19$ | $2.79 \pm 0.24$ | $0.90 \pm 0.00$ | $98.02 \pm 18.37$ | $2.46 \pm 0.16$ |
| | **0.25** | $1.00 \pm 0.00$ | $162.05 \pm 48.05$ | $2.89 \pm 0.34$ | $0.90 \pm 0.07$ | $99.51 \pm 18.53$ | $2.48 \pm 0.18$ |
| | **0.5** | $1.00 \pm 0.00$ | $175.54 \pm 62.29$ | $2.82 \pm 0.36$ | $0.90 \pm 0.11$ | $110.35 \pm 19.93$ | $2.56 \pm 0.20$ |
| | **1.0** | $0.99 \pm 0.02$ | $172.51 \pm 61.13$ | $2.96 \pm 0.40$ | $0.91 \pm 0.04$ | $106.63 \pm 28.84$ | $2.64 \pm 0.28$ |
| Hard | **None** | $0.99 \pm 0.09$ | $301.13 \pm 52.61$ | $3.34 \pm 0.58$ | $0.58 \pm 0.02$ | $262.23 \pm 45.36$ | $4.64 \pm 0.36$ |
| | **0.15** | $1.00 \pm 0.00$ | $308.22 \pm 38.40$ | $3.47 \pm 0.26$ | $0.63 \pm 0.04$ | $237.00 \pm 35.72$ | $4.06 \pm 0.31$ |
| | **0.25** | $1.00 \pm 0.02$ | $305.62 \pm 39.73$ | $3.32 \pm 0.35$ | $0.66 \pm 0.11$ | $238.92 \pm 28.66$ | $3.90 \pm 0.27$ |
| | **0.5** | $1.00 \pm 0.03$ | $314.36 \pm 37.93$ | $3.38 \pm 0.32$ | $0.59 \pm 0.14$ | $250.47 \pm 29.71$ | $4.16 \pm 0.24$ |
| | **1.0** | $0.99 \pm 0.03$ | $313.51 \pm 61.64$ | $3.69 \pm 0.44$ | $0.64 \pm 0.09$ | $256.24 \pm 12.04$ | $4.24 \pm 0.18$ |

`GiveMeSomeCredit`, we get $\approx 0.15$ (noiseless, $k = 4$) and $\approx 0.45$ (logistic, $k = 2$). Overall, we can provide interventions which are at least 50% cheaper than their preference-agnostic counterparts.

**Q2: `PEAR` outperforms competitors in terms of validity and cost.** Following the AR literature (Karimi et al., 2022; Verma et al., 2020), we compare `PEAR` (with $T = 10$) and all competitors in terms of average *validity*, *i.e.*, fraction of users for which we obtained recourse, intervention *cost* and *length* (or *sparsity*), *i.e.*, the number of features that have to be changed. Intervention costs are computed by using the true weights $\mathbf{w}^{GT}$. Table 1 shows the results. `PEAR` manages to achieve the highest validity while also providing substantially cheaper interventions than the non-personalized competitors on average. This is true both for the noiseless and logistic response models. While `CSCF` tends to produce shorter interventions, these are in general more costly and have a larger cost variance with respect to those found by `PEAR`, confirming the intuition that length is a suboptimal proxy of intervention complexity. The only exception is the `Hard` setting of `GiveMeSomeCredit`, where however `CSCF` manages to achieve recourse for only 22% of the users, whereas `PEAR` achieves recourse in 58% of the cases. The difficulty of `CSCF` in achieving recourse is visible in all settings and severely limits its applicability. Furthermore, `CSCF` is 10 to 50 times more computationally expensive than `PEAR`, making it unsuitable for real-time interactive scenarios. The `MCTS` baseline has rather poor performance both in terms of validity and cost in all settings, while the `RL` baseline has a reasonably high validity on `Adult` but it completely fails to learn a policy achieving recourse on `GiveMeSomeCredit`. On the other hand, methods which combine `MCTS` and `RL` (`FARE` and `EFARE`) give better performance, which is aligned with previous results (De Toni et al., 2023), but are still suboptimal with respect to `PEAR` in terms of both validity and cost. Finally, `FACE` struggles to achieve recourse since it needs to find a "feasible path" from the current user to a similar one *in the training set*, which is favourably classified.

**Q3: `PEAR` is robust to misspecifications of the cost correlation structure.** In the previous experiments, following other research works (Karimi et al., 2021; De Toni et al., 2023; Naumann & Ntoutsi, 2021), we assumed to know the structure of the CCS a-priori. However, in a real scenario, we might have instead an *approximate* causal graph from which to derive the CCS. Table 2 shows the validity, cost and length of the interventions found by removing $X\%$ of edges from the causal graph, with $X \in \{0.15, 0.25, 0.50, 1.00\}$. Validity is almost unaffected by corruption in all settings since it only impacts the computation of the cost. On the other hand, as expected, increasing the amount of graph corruption reduces the effectiveness of user feedback. However, the degradation is not dramatic. Indeed, if we look at the `Hard` evaluation, the increase in cost is negligible (around 4%) with up to 50% randomly removed edges. On `GiveMeSomeCredit`, we do not see any significant increase in costs. Surprisingly, we see instead an improvement for 15% and 25% corruption levels. We hypothesize that lacking causal knowledge about features which are not needed for recourse can be beneficial since it simplifies the elicitation process. However, at higher levels of corruption, this effect disappears. The setting $X = 1.0$ is equivalent to a non-causal cost function, in which acting on a feature has always the same cost, irrespective of the others. It is the common choice of many works dealing with

AR (Wachter et al., 2017; Ramakrishnan et al., 2020; Poyiadzi et al., 2020). Under such a setting, when considering `All` users, the degradation is more evident, but still within 6% for `GiveMeSomeCredit`, while for `Adult` it goes up to 15%. Overall, results clearly indicate that `PEAR` can suggest reasonable cost interventions even with a largely misspecified cost correlation structure. This is apparent when comparing these results with those in Table 1. Even with 50% randomly removed edges, `PEAR` recommends interventions that are cheaper than all competitors but `CSCF`, that however has a substantially lower validity.

**Further experimental evaluations.** We performed some additional experiments which explored the relationship between the cost and length of an intervention $I$ and validated the usage of the model score $h(\mathbf{x})$ as a proxy for identifying the difficulty of finding a recourse suggestion. We report our findings in Appendix E. Lastly, following the example of De Toni et al. (2023), we also provide an explainable version of our method, `XPEAR`, which complements interventions with Boolean explanations during the elicitation process by taking into consideration the user effort. We then compared `XPEAR` against the corresponding baselines `EFARE` (De Toni et al., 2023) and while it provides good performances at the expense of being more interpretable, it is still outperformed by `PEAR`. The results are presented in Appendix D.

## 6 Broader Impact

Algorithmic recourse focuses on developing methods to increase the user's agency in contesting automated decision systems outcomes and the fairness of the current machine learning models. As for all systems impacting humans, we need to consider the possible bad ethical ramifications of these technologies and potential mitigation strategies. In the context of personalized algorithmic recourse systems, we can identify two sources of algorithmic bias which could hinder the quality of our recommendations:

**(1) Different recourse suggestions when conditioned on a protected attribute**. Users who share a similar profile, but differ in some sensitive features (e.g., sex, age, etc.) might obtain different interventions $I$. We believe this might be the most important source of unfairness. In such a scenario, a method might successfully optimize both Eq. (1) and Eq. (2), but some categories of users will always get costlier interventions. However, this is an issue shared by almost the totality of methods in the AR literature.

**(2) Uninformative prior lacking information about sensitive categories**. The prior $P(\mathbf{w})$ used by the Bayesian procedure in `PEAR` might not describe the preferences of less-represented groups, thus making it more difficult to learn such cost functions. Critically, methods employing a population-level estimate of $\mathbf{w}$ or which relies on expert evaluation (Rawal & Lakkaraju, 2020) have the same limitation.

We believe issue (1) relates to the quality underlying recourse generator method (e.g., `W-FARE`, `CSCF`, `FACE`, etc.). A solution could be employing AR strategies that are specifically optimized to satisfy a notion of fairness (Gupta et al., 2019; Haldar et al., 2022; von Kügelgen et al., 2022). However, to the best of our knowledge, this is a little-explored area in the recourse literature which would require additional investigation beyond the scope of the present work. Regarding issue (2), we could imagine focusing our efforts on collecting additional data on sensitive groups to learn a more informative prior to running `PEAR`. Additionally, we could raise the iteration budget ($T$) to increase the number of pairwise constraints we apply to our MCMC procedure. However, it would also increase the cognitive burden for the user.

Lastly, eliciting users' preferences might entail asking sensitive questions, or malicious entities could exploit these procedures to "hack" and twist the intervention generation via strategic behaviour (Hardt et al., 2016). These considerations can be mitigated by research on adversarial attacks to ensure the method's robustness (Chen et al., 2020). Moreover, legal advice might be needed to manage personal user data.

## 7 Conclusion

In this work, we identify the problem of *personalized* algorithmic recourse as a fundamental stepping stone for ensuring recourse is usable in real-world applications, and develop `PEAR`, the first algorithm able to provide *personalized* interventions. Our experimental evaluation shows that `PEAR` substantially outperforms existing (non-personalized) solutions in terms of both validity and intervention cost with only a handful of queries to the user. We hope that this initial contribution can foster further research in the community to work towards

a more realistic form of algorithmic recourse that can be successfully deployed in real-world scenarios. As for all methods dealing with algorithmic recourse, the effectiveness of the approach should, in principle, be evaluated on real users. However, this evaluation is highly non-trivial (and thus still missing in the algorithmic recourse literature) because it requires the creation of a realistic scenario where a user feels to be *unfairly treated* in some machine-driven decision involving her life. The legal requirements that are progressively being introduced to regulate AI systems (Voigt & Von dem Bussche, 2017) could contribute to making the information needed to set up such a scenario available in the near future.

## Acknowledgments

Funded by the European Union. Views and opinions expressed are however those of the author(s) only and do not necessarily reflect those of the European Union or the European Health and Digital Executive Agency (HaDEA). Neither the European Union nor the granting authority can be held responsible for them. Grant Agreement no. 101120763 - TANGO. AP and BL also acknowledge the support of the MUR PNRR project FAIR - Future AI Research (PE00000013) funded by the NextGenerationEU. BL also acknowledges the support of the European Commission under Horizon Europe Programme, grant number 101120237 - ELIAS. The work of Giovanni De Toni was partially supported by the project AI@Trento (FBK-Unitn).

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

# A  Competitors and Experimental Settings

We now briefly illustrate the competitors' main principles and describe their most important hyperparameters. We also discuss `W-FARE` and its new reward function in more detail.

**Implementation details.** We implemented `PEAR`, the competitors and the black box classifiers using Python ($>= 3.7$) and PyTorch (Paszke et al., 2019). For reproducibility purposes, the code and the pre-trained models are freely available online[4]. We used the original code for both `FARE`[5] and `CSCF`[6], with minimal modifications to make them compatible with our experimental setting. For `FACE`, we used the implementation available in the CARLA library (Pawelczyk et al., 2021). As for the baselines, we adapted `MCTS` and the `RL` agent from the `FARE` repository. All the experiments were run on a virtual machine running CentOS 7.6.18 with 165 cores, and 25 GiB of RAM. The full set of hyperparameter configurations is available in the code.

**State space and Action space.** Given the raw datasets, we one-hot encoded categorical features and we performed min-max normalization for the continuous features using `scikit-learn` (Pedregosa et al., 2011). We also manually performed additional standard data engineering tasks, such as removing entries with null values or checking for potential outliers. After the data cleaning and preprocessing steps, we kept the following features for each dataset:

- `Adult`: *age*, *capital_gain*, *capital_loss*, *hours_per_week* (continuous) *workclass*, *education*, *marital_status*, *occupation*, *relationship*, *race*, *sex*, *native_country* (categorical). We considered the features *age*, *relationship*, *race*, *sex*, and *native_country* to be unactionable.

- `GiveMeSomeCredit`: *RevolvingUtilizationOfUnsecuredLines*, *age*, *NumberOfTime30-59DaysPastDueNotWorse*, *DebtRatio*, *MonthlyIncome*, *NumberOfOpenCreditLinesAndLoans*, *NumberOfTimes90DaysLate*, *NumberRealEstateLoansOrLines*,*NumberOfTime60-89DaysPastDueNotWorse*, *NumberOfDependents* (continuous). We considered the features *age* and *NumberOfDependents* to be unactionable.

These features represent the state $\mathbf{s} \in \mathcal{S}$ of a user. Table 3 reports the action space $\mathcal{A}$ used by `W-FARE` and the baselines for each of the experimental datasets. For `Adult`, we adopted the same action set used by De Toni et al. (2023), while for `GiveMeSomeCredit` we devised the functions ourselves. Please note that the actions $a \in \mathcal{A}$ operate directly in the original feature space, without encoding or rescaling, to ensure the interventions are understandable by humans. The features assigned as non-actionable do not have any action available. We refer the reader to the code implementation for additional details.

**Cost Correlation Structure.** Following the previous literature, (Karimi et al., 2021; De Toni et al., 2023; Naumann & Ntoutsi, 2021), we manually designed the directed acyclic graph for both experiments. Fig. 4 shows the resulting graphs. During the graph corruption experiments, we incrementally remove a subset of edges taken uniformly at random.

## A.1  Monte Carlo Tree Search (`MCTS`)

`MCTS` (Kocsis & Szepesvári, 2006; Coulom, 2007) is an efficient tree-based heuristic search procedure that can solve difficult tasks where computing directly the exact solution is intractable (Silver et al., 2016). `MCTS` builds a tree representing all possible moves and outcomes. Each tree node represents a state $\mathbf{s}^{(i)}$ and each edge represents performing an action $a \in \mathcal{A}$. In each iteration, `MCTS` builds this tree following four main steps: selection, expansion, simulation, and backpropagation. The *selection* step takes a state $\mathbf{s}^{(t)}$ and selects the next action $a^{(t+1)}$, based on some strategy such as UCB1 (Kocsis & Szepesvári, 2006). In our experiments, `MCTS` uses the P-UCT (Rosin, 2011; Silver et al., 2016) criterion, defined as:

$$a^{(t+1)} = \underset{a' \in \mathcal{A}}{\operatorname{argmax}} \ \mathbb{E}[r|\mathbf{s}^{(t)}, a'] + U(\mathbf{s}^{(t)}, a') + L(\mathbf{s}^{(t)}, a') \tag{12}$$

---

[4]https://github.com/unitn-sml/pear-personalized-algorithmic-recourse
[5]https://github.com/unitn-sml/syn-interventions-algorithmic-recourse
[6]https://github.com/ppnaumann/CSCF

Table 3: Action set $\mathcal{A}$ for `Adult` and `GiveMeSomeCredit` datasets used by `W-FARE` and the baselines. We report the functions, their arguments and which feature each of them modifies. For `Adult`, we adopted the same action set as De Toni et al. (2023), but we added two additional functions, CHANGE_CAP_LOSS and CHANGE_CAP_GAIN to increase the available interventions. We manually crafted the action space for `GiveMeSomeCredit`. In the case of continuous features, we constructed the argument sets by picking evenly spaced numbers from an interval since MCTS cannot handle continuous search spaces. We report the intervals from which we selected the argument values. For categorical features, we simply report the full set of options available.

### Adult

| Feature | Function ($f$) | Argument Type | Argument ($x$) |
|---|---|---|---|
| workclass | CHANGE_WORKCLASS | Categorical | Never-worked, Without-pay, Self-emp-not-inc, Self-emp-inc, Private, Local-gov, State-gov, Federal-gov |
| education | CHANGE_EDUCATION | Categorical | Preschool, 1st-4th, 5th-6th, 7th-8th, 9th, 10th, 11th, 12th, HS-grad, Some-college, Bachelors, Masters, Doctorate, Assoc-acdm, Assoc-voc, Prof-school |
| occupation | CHANGE_OCCUPATION | Categorical | Tech-support, Craft-repair, Other-service, Sales, Exec-managerial, Prof-specialty, Handlers-cleaners, Machine-op-inspct, Adm-clerical, Farming-fishing, Transport-moving, Priv-house-serv, Protective-serv, Armed-Forces |
| capital_gain | CHANGE_CAP_GAIN | Integer | $[-5000, -1] \cup [1, 5000]$ |
| capital_loss | CHANGE_CAP_LOSS | Integer | $[-5000, -1] \cup [1, 5000]$ |
| hours_per_week | CHANGE_HOURS | Integer | $[-25, -1] \cup [1, 25]$ |

### GiveMeSomeCredit

| Feature | Function ($f$) | Argument Type | Argument (x) |
|---|---|---|---|
| RevolvingUtilizationOfUnsecuredLines | BALANCE | Real | $[-1, -0.01] \cup [0.01, 1]$ |
| NumberOfTime30-59DaysPastDueNotWorse | PAST_DUE_30 | Integer | $[-25, -1]$ |
| DebtRatio | DEBT_RATIO | Real | $[-1, -0.01] \cup [0.01, 1]$ |
| MonthlyIncome | INCOME | Integer | $[-10000, -1] \cup [1, 10000]$ |
| NumberOfOpenCreditLinesAndLoans | OPEN_LOANS | Integer | $[-25, -1]$ |
| NumberOfTimes90DaysLate | PAST_DUE_90 | Integer | $[-25, -1]$ |
| NumberRealEstateLoansOrLines | OPEN_REAL_ESTATE | Integer | $[-25, -1]$ |
| NumberOfTime60-89DaysPastDueNotWorse | PAST_DUE_60 | Integer | $[-25, -1]$ |

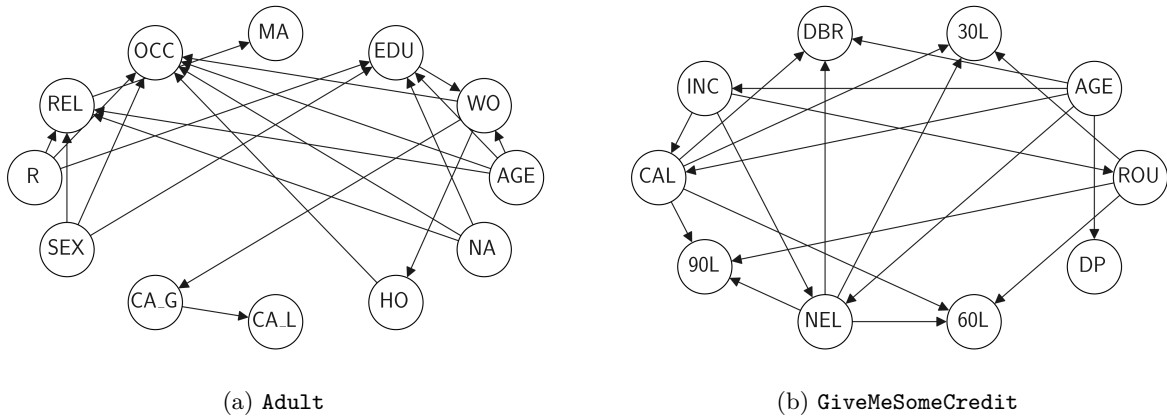

(a) `Adult`

(b) `GiveMeSomeCredit`

Figure 4: **Directed Acyclic Graphs used for the Cost Correlation Structures.** Each node corresponds to a specific user feature. The complete mapping between node acronyms and features can be found in the experimental configuration files.

Here, $\mathbb{E}[r|\mathbf{s}^t, a']$ is the expected reward of taking action $a'$ in state $\mathbf{s}^t$. The reward $r$ is $\gamma^{(|I|)}$ if we achieve recourse and $-1$ otherwise. $|I|$ is the length of the intervention while $\gamma \in (0, 1]$ is a discount factor. $L(\mathbf{s}^{(t)}, a')$ is a penalty is equal to $c_{act\_cost} \exp(-C(a', \mathbf{s}^{(t)} \mid \mathbf{w}))$. $U(\mathbf{s}^{(t)}, a')$ trades off exploration and exploitation, and it is defined as

$$U(\mathbf{s}^{(t)}, a) = c_{puct} \cdot P(a \mid \mathbf{s}^{(t)}) \frac{\sqrt{N_{\mathbf{s}^{(t)}}}}{1 + N_{\mathbf{s}^{(t+1)}}} \tag{13}$$

where $P(a \mid \mathbf{s}^{(t)})$ is the prior probability of choosing such action, $N_{\mathbf{s}^{(t)}}$ is the visit count of node $\mathbf{s}^{(t)}$, and $N_{\mathbf{s}^{(t+1)}}$ is the visit count of the child node being evaluated. In our experiments, the prior $P(a \mid \mathbf{s}^{(t)})$ is a uniform distribution over the actions.

The *expansion* step adds new child nodes to the current node based on the available actions. The *simulation* step then carries out a *playout*, selecting moves randomly until a terminal state is reached. The value of the outcome of this terminal state is then backpropagated up the tree (*backpropagation* step), incrementing the visit $N_{\mathbf{s}^{(i)}}$ and reward counts of nodes along the path that was traversed to reach the outcome. We use the reward counts to compute $\mathbb{E}[r|\mathbf{s}^{(t)}, a']$. The *number of simulations* is a hyperparameter. Once the simulations are done, we start over from the *selection* step. `MCTS` keeps expanding and selecting nodes until we reach recourse, or a user-defined *maximum intervention length*. Once `MCTS` ends, we return the successful intervention to the user. An intervention here is a *path* from the root node ($\mathbf{s}^{(0)}$) to a terminal node ($\mathbf{s}^{(T)}$).

**Hyperparameters.** In our experiments, we set the *number of simulations* to 15 and 10 for `Adult` and `GiveMeSomeCredit`, respectively. We also set the *maximum intervention length* to 6 and 8, for `Adult` and `GiveMeSomeCredit`, respectively. The value $c_{act\_cost}$ and $c_{puct}$ are also hyperparameters. We set them to 1 and 0.5 respectively, for both experiments.

### A.2 Recourse Policy (`RL`)

`RL` is a simple agent which learns a recourse policy. Following the architecture described in De Toni et al. (2023), it has four different components: an encoder $g_{enc}$, an LSTM controller $f_c$, a function network $g_f$, an argument network $g_x$ and a value network $g_V$. $g_{enc}$ produces a latent representation of the current state $\mathbf{s}^{(t)}$ which is sent to the controller $f_c$. Given the hidden states $h_t$ coming from $f_c$, the function and argument network generates the policies $\pi_f$ and $\pi_x$ which we use to get the tuple $(f, x)^{(t+1)}$ which corresponds to the next action $a$. We use the value network $g_V$ to approximate the value function $V(\mathbf{s}^{(t)}, a)$.

**Training the agent.** We train `RL` such to minimize the following loss function

$$\ell = \sum_{batch} (V - r^2) - \pi_f^{true\,T} \log(\pi_f) - \pi_x^{true\,T} \log(\pi_x) \tag{14}$$

where we assume to have access to the ground truth policies $\pi_f^{true}$ and $\pi_x^{true}$ either by having a previous dataset of recourse options, by optimizing over an MDP following (Verma et al., 2022), or by discovering such traces by direct search. In our paper, we follow the last approach by using `MCTS` as the search component. At inference time, we run the agent until we either find recourse, reach the *maximum intervention length*, or call an illegal action (e.g., an action whose *precondition* is not satisfied).

**Hyperparameters.** As in De Toni et al. (2023), $g_{enc}$, $g_f$ and $g_x$ are all MLPs and we manually defined their structure for each experimental setting (e.g., number of layers, dimensions, etc.). We do the same for the controller $f_c$. We optimize Eq. (14) via Adam and we set the *learning rate* to 0.001 for `Adult`, and 0.003 for `GiveMeSomeCredit`.

### A.3 (Explainable) eFficient counterfActual REcourse (`FARE` and `EFARE`)

`FARE` (De Toni et al., 2023) is a model-agnostic method which exploits the synergy between a learned (`RL`) policy and a discrete search procedure (`MCTS`) to discover recourse options efficiently. At training time, differently from `MCTS` and `RL`, we do not use a uniform prior, but we leverage the agent instead $P(a \mid \mathbf{s}^{(t)}) = \pi_f(\mathbf{s}^{(t)}) \cdot \pi_x(\mathbf{s}^{(t)})$. Similarly to Silver et al. (2016), we foster additional exploration by replacing $P(a \mid \mathbf{s}^{(t)})$ with a linear combination, $\epsilon_P P(a \mid \mathbf{s}^{(t)}) + (1 - \epsilon_P)\eta_P$ where $\eta_P \sim \text{Dir}(\alpha_d)$. We manually set the parameters $\alpha_d$ and $\epsilon_P$ for each setting.

`EFARE` is a deterministic model which provides sequential counterfactuals together with explanations. Given a trained `FARE` model, a sample of successful interventions for the training set users is extracted. These *interventions* are merged to form a graph, where each node represents an action $a$, and each arch represents which actions $(f, x)$ can be taken from the said node. Then, for each node, a small decision tree is trained to indicate, given a user state $\mathbf{s}^{(t)}$, which is the next recommended action. Given this trained automaton, we can traverse the graph using the decision trees to transition between nodes. Moreover, the decision trees can give us Boolean explanations for each transition.

**Hyperparameters.** `FARE` shares the same hyperparameters of `MCTS` and `RL`. During training, we set the *number of simulations* to 15 and 10, for `Adult` and `GiveMeSomeCredit`, respectively. The noise fraction is instead set to $\epsilon_P = 0.3$ for both, with $\eta_p = 0.3$. At inference time, we add no noise ($\epsilon_P = 0$) and the *number of simulations* is fixed at 5. Lastly, for each dataset, we train a corresponding `EFARE` version by sampling 300 interventions from the training set using `FARE`.

### A.4 Consequence-aware Sequential Counterfactuals (`CSCF`)

`CSCF` (Naumann & Ntoutsi, 2021) is a model-agnostic evolutionary algorithm which can generate sequential recourse plans by taking into consideration feature relationships. `CSCF` uses the Biased Random-Key Genetic Algorithm (BRKGA) (Gonçalves & Resende, 2011) together with non-dominated sorting (NDS) (Srinivas & Deb, 1994) to perform multi-objective optimization. BRKGA optimizes over the solution *genotype* (real-values vectors $G \in [0, 1]$) and evaluates its *phenotype* instead (sequences of actions $a \in \mathcal{A}$). Thus, they propose also a *decoder* to deterministically derive a single phenotype from a genotype.

At each iteration, `CSCF` evaluates the decoded solutions phenotypes by computing their *fitness*. We use the same fitness function as proposed in the original paper. Then, following NDS, a new population is formed by genetic mating and biased crossover between the *elites* (feasible and valid solutions in the Pareto front) and *non-elites*. The loop then continues until we reach a termination condition (e.g., we reach the maximum number of generations). From the last population set, we pick the lowest cost valid intervention with ties broken uniformly at random.

**Hyperparameters.** In our experiments, we mainly used the default hyperparameters provided by the authors (Naumann & Ntoutsi, 2021). We set the population size, $p = 50$, and the maximum number of generations, $n = 25$, for both `Adult` and `GiveMeSomeCredit`, to keep the computation time manageable. Higher values might improve the performances, but it would make `CSCF` unusable in an online setting (e.g., web interface provided by the bank).

### A.5 Feasible and Actionable Counterfactual Explanations (FACE)

FACE ([Poyiadzi et al., 2020](#)) is a model-agnostic method which can find actionable counterfactual examples respecting the underlying data distribution. The main idea is to propose recourse options based on the shortest path between the current state of the user and a state reaching recourse. FACE proposes different density-weighted metrics to find these "feasible paths". FACE starts by building a graph over the training set data points. Then, the user can specify certain properties (e.g., prediction threshold, custom weight function, etc.), which restrict the graph by removing edges pointing to unfeasible counterfactuals. FACE then uses the Shortest Path First Algorithm (Dijkstra's algorithm) over all the instances matching the requirements in the graph.

We use FACE based on the proposed $k$-NN construction algorithm. Since FACE provides *counterfactual examples* and not sequences, we then return the lowest-cost counterfactual by applying all the changes sequentially. The action order is randomly chosen.

**Hyperparameters.** CARLA's implementation of FACE requires computing an adjacency matrix given the instances in the training set. The procedure is quite expensive since it requires $O(n^2)$ memory, where $n$ is the number of instances. Thus, at inference time, we sample only a fraction of the original dataset. In both Adult and GiveMeCredit, we pick only 10% of the total instances. We set the *number of neighbours*, $k$, to 50 and the distance threshold to $\epsilon = 1.0$ for both datasets.

## B   Weighted-FARE (W-FARE)

FARE is user-agnostic since it is trained by assuming a shared $\mathbf{w}$ for all instances. Moreover, its reward function optimizes only for the length of the intervention, without explicating the cost $C(I \mid \mathbf{w})$. In our experiments, we train FARE with $\mathbb{E}_{P(\mathbf{w})}[\mathbf{w}]$. Thus, we push FARE to learn a general policy which works best in the average case, rather than for a specific user. To overcome this limitation, W-FARE proposes a new reward function which optimizes for the optimal intervention and tailors its answers to a given set of preferences $\mathbf{w}$. The new reward is computed as

$$r = \begin{cases} \gamma^{|I|+C(I|\mathbf{w})} \cdot \sigma\left(C(I_{\mathbb{E}_{P(\mathbf{w})}[\mathbf{w}]} \mid \mathbf{w}) - C(I \mid \mathbf{w})\right) & h(\mathbf{s}^{(0)}) \neq h(I(\mathbf{s}^{(0)})) \\ -1 & \text{otherwise} \end{cases} \tag{15}$$

where $\sigma(x) = \frac{1}{1+\exp(-x\eta)}$ and $\eta = 0.01$ and $\gamma = 0.97$. The first term $\gamma^{|I|+C(I|\mathbf{w})}$ optimizes for short and cheap interventions, while the second term $\sigma(C(I_{\mathbb{E}_{P(\mathbf{w})}[\mathbf{w}]} \mid \mathbf{w}) - C(I \mid \mathbf{w}))$ encourages W-FARE to provide *personalized* recourse options, by generating sequences which have lower costs than the intervention found by assuming the expected value $\mathbb{E}_{P(\mathbf{w})}[\mathbf{w}]$. We also set the $L(\mathbf{s}^{(t)}, a')$ term of the MCTS's P-UCT criterion to $c_{act\_cost} \cdot \gamma^{|I|+C(I|\mathbf{w})} \cdot \sigma(C(I_{\mathbb{E}_{P(\mathbf{w})}[\mathbf{w}]} \mid \mathbf{w}) - C(I \mid \mathbf{w}))$ to help the model converge.

In both RL and FARE, the user preferences are not visible in the state $\mathbf{s}$. Given a user state $\mathbf{s}^{(t)}$, W-FARE adds the cost of every valid action $a \in \mathcal{A}$ as an additional feature. Since an action $a = (f, x)$ is a combination of a function $f$ and an argument $x$, we combine actions relating to the same function into a single feature, averaging their cost over the set of argument values. This simplification allows adding $|\mathcal{F}|$ additional features to the user state, where $\mathcal{F}$ is the set of all possible functions. More formally, the user's state $\mathbf{s}'$ becomes

$$\mathbf{s}' = \mathbf{s} \circ \left\{ \frac{1}{|\mathcal{X}_f|} \sum_{x \in \mathcal{X}_f} C((f, x), \mathbf{s} \mid \mathbf{w}) \right\}_{f \in \mathcal{F}} \tag{16}$$

where $\mathcal{X}_f$ is the set of all the possible arguments for function $f$. We also considered adding directly the weights $\mathbf{w}$, but we discovered that such representation makes it harder to learn an optimal policy. Ultimately, we train W-FARE by assigning to each user in the training set a dummy weight vector sampled from the prior $\mathbf{w} \sim P(\mathbf{w})$.

## C   Evaluating PEAR on shallow classifiers

We evaluated the performances of PEAR also by using non-neural machine learning models. Namely, we trained a logistic regression and a tree-based classifier. We performed hyperparameter optimization using grid

Table 4: Performance of all competitors averaged over 10 runs. A '-' indicates that the method did not find *any* successful intervention for *any* user. PEAR$_{NL}$ and PEAR$_L$ indicate PEAR associated with the noiseless and logistic response model, respectively. Please note that while some competitors can sometimes achieve a slightly lower cost compared to PEAR, they suffer from much worse validity.

**Logistic Regression**

| Users | Corruption | Adult | | | GiveMeSomeCredit | | |
|---|---|---|---|---|---|---|---|
| | | *Validity* | *Cost* | *Length* | *Validity* | *Cost* | *Length* |
| All | FARE | $0.97 \pm 0.16$ | $266.48 \pm 166.07$ | $3.44 \pm 0.87$ | $0.89 \pm 0.18$ | $156.40 \pm 88.92$ | $3.15 \pm 0.92$ |
| | EFARE | $0.79 \pm 0.35$ | $296.51 \pm 182.87$ | $3.59 \pm 0.98$ | $0.72 \pm 0.34$ | $161.22 \pm 90.93$ | $3.18 \pm 0.95$ |
| | RL | $0.78 \pm 0.37$ | $268.07 \pm 170.85$ | $3.43 \pm 0.94$ | $0.15 \pm 0.36$ | $58.43 \pm 8.93$ | $2.00 \pm 0.00$ |
| | CSCF | $0.73 \pm 0.26$ | $167.41 \pm 94.55$ | $2.81 \pm 0.44$ | $0.61 \pm 0.39$ | $109.48 \pm 113.21$ | $2.54 \pm 1.04$ |
| | MCTS | $0.45 \pm 0.44$ | $435.23 \pm 206.31$ | $4.55 \pm 1.17$ | $0.72 \pm 0.41$ | $214.38 \pm 109.90$ | $4.02 \pm 1.22$ |
| | FACE | $0.21 \pm 0.20$ | $419.92 \pm 97.07$ | $3.92 \pm 0.45$ | $0.37 \pm 0.36$ | $319.30 \pm 48.95$ | $5.79 \pm 0.49$ |
| | PEAR$_{NL}$ (ours) | $\mathbf{0.99 \pm 0.09}$ | $\mathbf{146.28 \pm 64.08}$ | $\mathbf{2.96 \pm 0.31}$ | $\mathbf{0.95 \pm 0.07}$ | $\mathbf{90.07 \pm 22.64}$ | $\mathbf{2.50 \pm 0.25}$ |
| | PEAR$_L$ (ours) | $\mathbf{0.99 \pm 0.09}$ | $\mathbf{154.29 \pm 66.55}$ | $\mathbf{2.98 \pm 0.38}$ | $\mathbf{0.95 \pm 0.08}$ | $\mathbf{90.88 \pm 28.86}$ | $\mathbf{2.49 \pm 0.28}$ |
| Hard | FARE | $0.90 \pm 0.28$ | $471.54 \pm 167.12$ | $4.83 \pm 0.97$ | $0.46 \pm 0.32$ | $332.35 \pm 81.30$ | $5.02 \pm 0.75$ |
| | EFARE | $0.59 \pm 0.45$ | $484.60 \pm 171.43$ | $4.84 \pm 0.96$ | $0.25 \pm 0.36$ | $346.12 \pm 92.08$ | $5.16 \pm 0.82$ |
| | RL | $0.56 \pm 0.47$ | $466.73 \pm 175.00$ | $4.74 \pm 1.04$ | $0.00 \pm 0.05$ | $83.53 \pm 0.00$ | $2.00 \pm 0.00$ |
| | CSCF | $0.22 \pm 0.34$ | $395.89 \pm 124.63$ | $4.20 \pm 0.67$ | $0.12 \pm 0.30$ | $190.70 \pm 110.23$ | $3.40 \pm 0.94$ |
| | MCTS | $0.20 \pm 0.38$ | $583.91 \pm 119.72$ | $5.56 \pm 0.59$ | $0.40 \pm 0.43$ | $348.86 \pm 94.76$ | $5.36 \pm 0.94$ |
| | FACE | $0.02 \pm 0.10$ | $598.63 \pm 11.96$ | $4.88 \pm 0.15$ | $0.29 \pm 0.36$ | $448.70 \pm 39.45$ | $6.96 \pm 0.49$ |
| | PEAR$_{NL}$ (ours) | $\mathbf{0.95 \pm 0.16}$ | $\mathbf{320.97 \pm 89.93}$ | $\mathbf{4.23 \pm 0.36}$ | $\mathbf{0.73 \pm 0.16}$ | $\mathbf{259.74 \pm 45.29}$ | $\mathbf{4.28 \pm 0.46}$ |
| | PEAR$_L$ (ours) | $\mathbf{0.96 \pm 0.15}$ | $\mathbf{329.40 \pm 93.83}$ | $\mathbf{4.27 \pm 0.42}$ | $\mathbf{0.74 \pm 0.17}$ | $\mathbf{261.38 \pm 45.67}$ | $\mathbf{4.28 \pm 0.47}$ |

**Decision Tree**

| Users | Corruption | Adult | | | GiveMeSomeCredit | | |
|---|---|---|---|---|---|---|---|
| | | *Validity* | *Cost* | *Length* | *Validity* | *Cost* | *Length* |
| All | FARE | $\mathbf{0.94 \pm 0.15}$ | $\mathbf{142.19 \pm 80.14}$ | $\mathbf{2.33 \pm 0.39}$ | $0.67 \pm 0.40$ | $129.17 \pm 148.29$ | $2.70 \pm 1.18$ |
| | EFARE | $0.71 \pm 0.22$ | $261.22 \pm 188.57$ | $3.18 \pm 0.98$ | $0.81 \pm 0.31$ | $213.55 \pm 105.56$ | $3.93 \pm 1.09$ |
| | RL | $0.70 \pm 0.31$ | $288.06 \pm 198.03$ | $3.37 \pm 1.13$ | $0.64 \pm 0.39$ | $235.94 \pm 109.32$ | $4.14 \pm 1.16$ |
| | CSCF | $0.55 \pm 0.31$ | $294.45 \pm 200.01$ | $3.43 \pm 1.07$ | $0.07 \pm 0.25$ | $96.50 \pm 105.43$ | $2.65 \pm 0.81$ |
| | MCTS | $0.47 \pm 0.46$ | $452.30 \pm 225.37$ | $4.50 \pm 1.31$ | $0.73 \pm 0.42$ | $229.19 \pm 122.03$ | $4.13 \pm 1.32$ |
| | FACE | $0.23 \pm 0.25$ | $450.83 \pm 112.93$ | $4.05 \pm 0.57$ | $0.59 \pm 0.39$ | $328.81 \pm 55.37$ | $5.72 \pm 0.58$ |
| | PEAR$_{NL}$ (ours) | $0.88 \pm 0.14$ | $169.03 \pm 64.91$ | $2.64 \pm 0.36$ | $\mathbf{0.95 \pm 0.11}$ | $\mathbf{113.21 \pm 29.33}$ | $\mathbf{2.70 \pm 0.28}$ |
| | PEAR$_L$ (ours) | $0.88 \pm 0.13$ | $168.41 \pm 57.40$ | $2.65 \pm 0.38$ | $\mathbf{0.95 \pm 0.11}$ | $\mathbf{113.47 \pm 31.94}$ | $\mathbf{2.69 \pm 0.31}$ |
| Hard | FARE | $\mathbf{0.92 \pm 0.14}$ | $\mathbf{117.19 \pm 54.11}$ | $\mathbf{2.32 \pm 0.34}$ | $0.44 \pm 0.43$ | $171.40 \pm 153.66$ | $3.09 \pm 1.27$ |
| | EFARE | $0.80 \pm 0.18$ | $226.16 \pm 191.83$ | $3.12 \pm 1.05$ | $0.56 \pm 0.39$ | $270.40 \pm 126.42$ | $4.44 \pm 1.16$ |
| | RL | $0.76 \pm 0.24$ | $233.65 \pm 197.26$ | $3.19 \pm 1.06$ | $0.35 \pm 0.41$ | $318.28 \pm 132.66$ | $4.81 \pm 1.13$ |
| | CSCF | $0.62 \pm 0.29$ | $229.10 \pm 198.01$ | $3.07 \pm 1.04$ | $0.02 \pm 0.15$ | $129.69 \pm 23.40$ | $2.68 \pm 0.45$ |
| | MCTS | $0.44 \pm 0.46$ | $424.77 \pm 228.23$ | $4.52 \pm 1.31$ | $0.57 \pm 0.45$ | $300.49 \pm 134.82$ | $4.77 \pm 1.27$ |
| | FACE | $0.08 \pm 0.20$ | $463.89 \pm 86.34$ | $4.36 \pm 0.51$ | $0.49 \pm 0.41$ | $457.94 \pm 57.34$ | $6.65 \pm 0.58$ |
| | PEAR$_{NL}$ (ours) | $0.84 \pm 0.12$ | $123.20 \pm 44.21$ | $2.50 \pm 0.30$ | $\mathbf{0.83 \pm 0.21}$ | $\mathbf{174.23 \pm 41.69}$ | $\mathbf{3.44 \pm 0.40}$ |
| | PEAR$_L$ (ours) | $0.84 \pm 0.12$ | $124.37 \pm 50.78$ | $2.50 \pm 0.33$ | $\mathbf{0.84 \pm 0.20}$ | $\mathbf{175.30 \pm 42.79}$ | $\mathbf{3.47 \pm 0.44}$ |

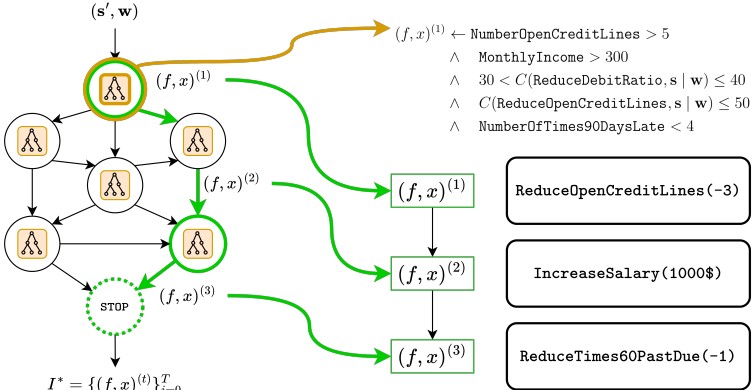

Figure 5: **The `W-EFARE` method.** Given $(\mathbf{s}', \mathbf{w})$, we traverse the graph (green path) using the decision tree (orange components) to choose the next action $(f, x)^{(t)}$ from the available transitions. When we reach the `STOP` node, we return the found intervention $I^*$. On the right, we have a generated intervention achieving recourse for the `GiveMeSomeCredit` dataset. On the top-right, we have the Boolean rule extracted from the decision tree motivating the first recommended action `ReduceOpenCreditLines`. As we can see, `W-EFARE` can generate rules that also depend on the cost function $C$, achieving the desired user awareness.

search *e.g.*, the *splitting criterion* (decision tree) and strength for the regularizer (logistic model). We then replicated the experiments described in Section 5 following the same protocol. In the case of the decision tree, the score of the classifier $h(\mathbf{x})$ is represented by the fraction of the examples of the same class in a leaf. Table 4 shows the evaluation results. `PEAR` tends to outperform the competitors in terms of cost and validity in 6 out of 6 (`GiveMeSomeCredit`) and 4 out of 6 (`Adult`) experiments. The only exception is decision trees on `Adult`, where `PEAR` is the runner-up behind `CSCF`, and still outperforms all other competitors.

## D Explaining Recommended Interventions with `XPEAR`

`PEAR` asks the users to choose the best option from a small set of interventions. The task of picking the best intervention requires cognitive effort since the user must understand all the recommended sequences of actions. This can be problematic if the user does not comprehend the reason behind the recommendations, undermining the quality of the feedback they provide and their overall trust in the system. To overcome this problem, we additionally developed a variant of `PEAR` that can complement interventions with action-specific explanations.

As detailed in Appendix A.3, `EFARE` (De Toni et al., 2023) is a deterministic user-agnostic model for generating recourse by providing Boolean explanations for each action in the intervention. The procedure can be adapted to account for user-specific costs. The resulting model, which we call `W-EFARE`, is a user-aware version of `EFARE` that can provide *personalized* explanations complementing the interventions. Fig. 5 shows a graphical illustration of how `W-EFARE` works in practice. We use this method to develop `XPEAR`, an explainable version of `PEAR`, which uses `W-EFARE` to generate both interventions and explanations which can be shown to the user in the choice set $O$.

Table 5 shows the evaluation results for `XPEAR`. We compare it against `PEAR`, `FARE` and `EFARE` since they share similar characteristics and assumptions. For the `Adult` dataset, `XPEAR` provides cheaper recourse options than the competitors (`FARE` and `EFARE`) by keeping a comparable or better validity. The behaviour is consistent for both `All` and `Hard` users. For the `GiveMeSomeCredit` dataset, `XPEAR` generates sequences with a better or similar validity and cost than the competitors, although the effect is diminished. In both datasets, `XPEAR` has higher validity than its deterministic counterpart `EFARE`. Overall, `PEAR` maintains the upper hand. However, it is worth noticing that `XPEAR` provides additional feedback to the users, which might be a valuable addition during the elicitation process.

Table 5: Performance of `XPEAR` averaged over 10 runs. $\text{XPEAR}_{NL}$ and $\text{XPEAR}_{L}$ indicate `XPEAR` associated with the noiseless and logistic response model, respectively. The best results are boldfaced.

| Users | Corruption | Adult | | | GiveMeSomeCredit | | |
|---|---|---|---|---|---|---|---|
| | | *Validity* | *Cost* | *Length* | *Validity* | *Cost* | *Length* |
| All | FARE | $0.90 \pm 0.27$ | $285.63 \pm 195.68$ | $3.45 \pm 1.19$ | $0.86 \pm 0.23$ | $161.77 \pm 107.42$ | $3.27 \pm 1.13$ |
| | EFARE | $0.76 \pm 0.39$ | $306.11 \pm 199.02$ | $3.54 \pm 1.25$ | $0.67 \pm 0.38$ | $155.92 \pm 109.19$ | $3.18 \pm 1.18$ |
| | $\text{PEAR}_{NL}$ | $\mathbf{1.00 \pm 0.03}$ | $\mathbf{142.23 \pm 61.75}$ | $\mathbf{2.84 \pm 0.59}$ | $\mathbf{0.89 \pm 0.00}$ | $\mathbf{96.04 \pm 31.96}$ | $\mathbf{2.79 \pm 0.42}$ |
| | $\text{PEAR}_{L}$ | $\mathbf{1.00 \pm 0.04}$ | $\mathbf{146.54 \pm 63.09}$ | $\mathbf{2.84 \pm 0.56}$ | $\mathbf{0.89 \pm 0.00}$ | $100.19 \pm 29.53$ | $\mathbf{2.85 \pm 0.48}$ |
| | $\text{XPEAR}_{NL}$ | $0.97 \pm 0.13$ | $181.28 \pm 96.83$ | $2.97 \pm 0.73$ | $0.83 \pm 0.18$ | $125.00 \pm 44.88$ | $2.84 \pm 0.40$ |
| | $\text{XPEAR}_{L}$ | $0.97 \pm 0.13$ | $203.08 \pm 112.29$ | $2.99 \pm 0.78$ | $0.85 \pm 0.19$ | $128.01 \pm 50.77$ | $2.84 \pm 0.46$ |
| Hard | FARE | $0.71 \pm 0.44$ | $438.97 \pm 188.50$ | $4.54 \pm 1.21$ | $0.47 \pm 0.37$ | $319.46 \pm 96.36$ | $4.97 \pm 0.70$ |
| | EFARE | $0.55 \pm 0.48$ | $454.05 \pm 202.76$ | $4.52 \pm 1.25$ | $0.22 \pm 0.36$ | $371.58 \pm 82.18$ | $5.31 \pm 0.71$ |
| | $\text{PEAR}_{NL}$ | $\mathbf{0.99 \pm 0.08}$ | $\mathbf{296.37 \pm 43.84}$ | $\mathbf{3.35 \pm 0.55}$ | $\mathbf{0.58 \pm 0.04}$ | $\mathbf{251.60 \pm 51.16}$ | $\mathbf{4.59 \pm 0.40}$ |
| | $\text{PEAR}_{L}$ | $\mathbf{0.99 \pm 0.09}$ | $\mathbf{301.13 \pm 52.61}$ | $\mathbf{3.34 \pm 0.58}$ | $\mathbf{0.58 \pm 0.02}$ | $\mathbf{262.23 \pm 45.36}$ | $\mathbf{4.64 \pm 0.36}$ |
| | $\text{XPEAR}_{NL}$ | $0.97 \pm 0.13$ | $326.68 \pm 98.44$ | $3.46 \pm 0.66$ | $0.44 \pm 0.30$ | $300.97 \pm 62.14$ | $4.81 \pm 0.50$ |
| | $\text{XPEAR}_{L}$ | $0.97 \pm 0.13$ | $334.58 \pm 96.28$ | $3.46 \pm 0.66$ | $0.45 \pm 0.31$ | $313.95 \pm 66.94$ | $4.88 \pm 0.56$ |

## E    Further evaluations of `PEAR`

We now present some graphs showing the relationship between the cost and length of the interventions (Fig. 6) and between the black box model score and the validity/interventions length (Fig. 7).

In Fig. 6, we can see that the length of an intervention can be a bad proxy for the intervention's difficulty. Namely, there are long interventions which still retain a small cost or effort for the user. Thus, we need to optimize for both the length and cost of interventions simultaneously. This notion is reflected in the reward function of `W-FARE` as we mentioned in Appendix B.

In Fig. 7, we see how the black box prediction score is a good metric to understand how hard it is for a user to obtain recourse. Namely, we suggest longer interventions for users obtaining a low score with respect to the others. Moreover, all the methods tend to fail to provide recourse to users with low scores. Interestingly, methods which learn a recourse policy rather than performing direct optimization (e.g., `PEAR` and `FARE`) are more capable of providing recourse generalizing over the low-score users.

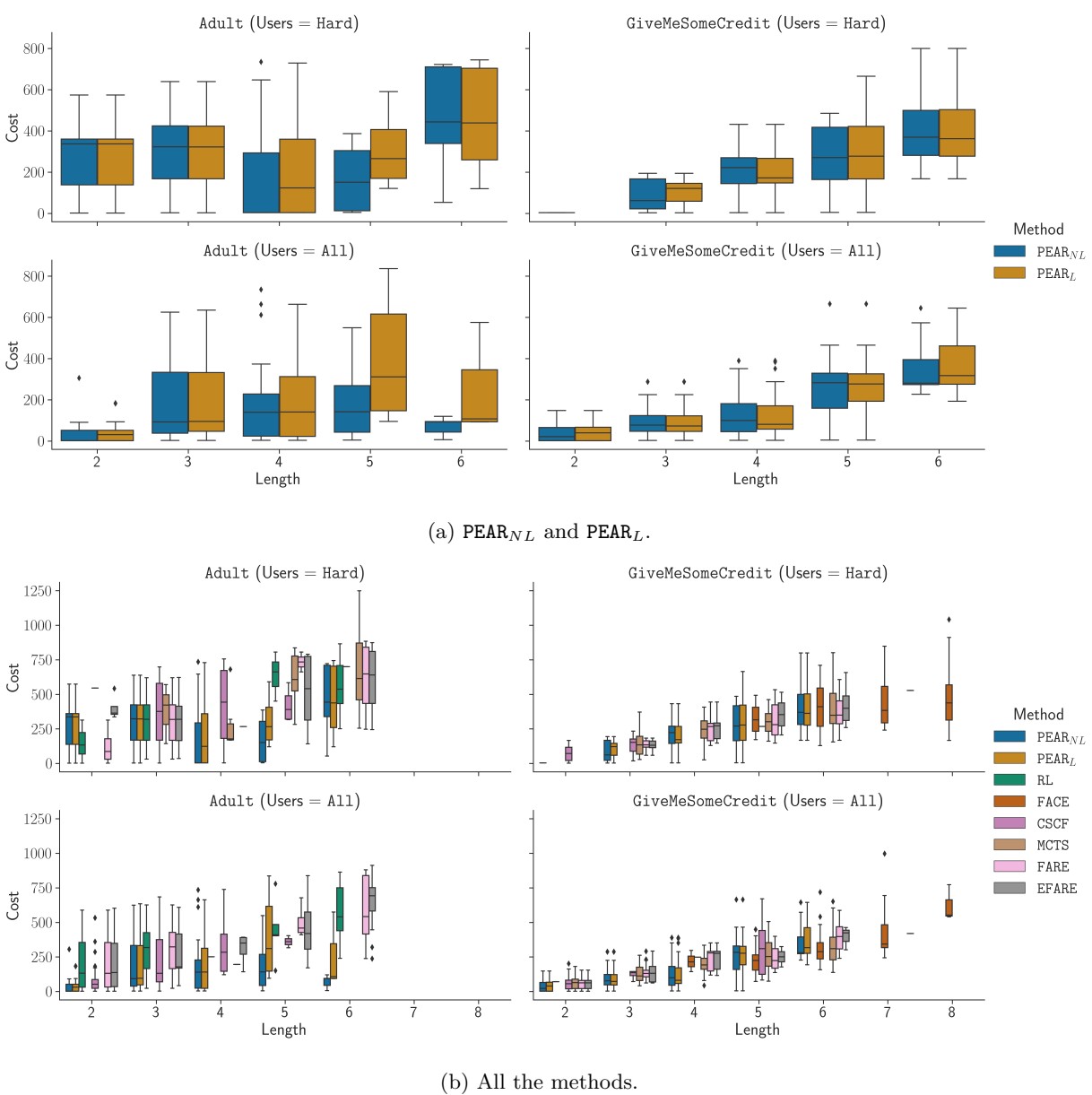

Figure 6: **Cost vs Intervention Length.** (Top) In the `Adult` dataset, `PEAR` generates interventions whose costs do not depend on their length (for both `All` and `Hard` users). The relationship is more pronounced on the `GiveMeSomeCredit` dataset, but longer interventions (e.g., $|I| = 5$) still present high variability in the costs. These results hint at the fact that length alone can be a bad proxy to measure the "effort" of a user. (Bottom) We plot the same results for all the methods. The relationship between cost and length is well represented here, and it aligns with the previous findings.

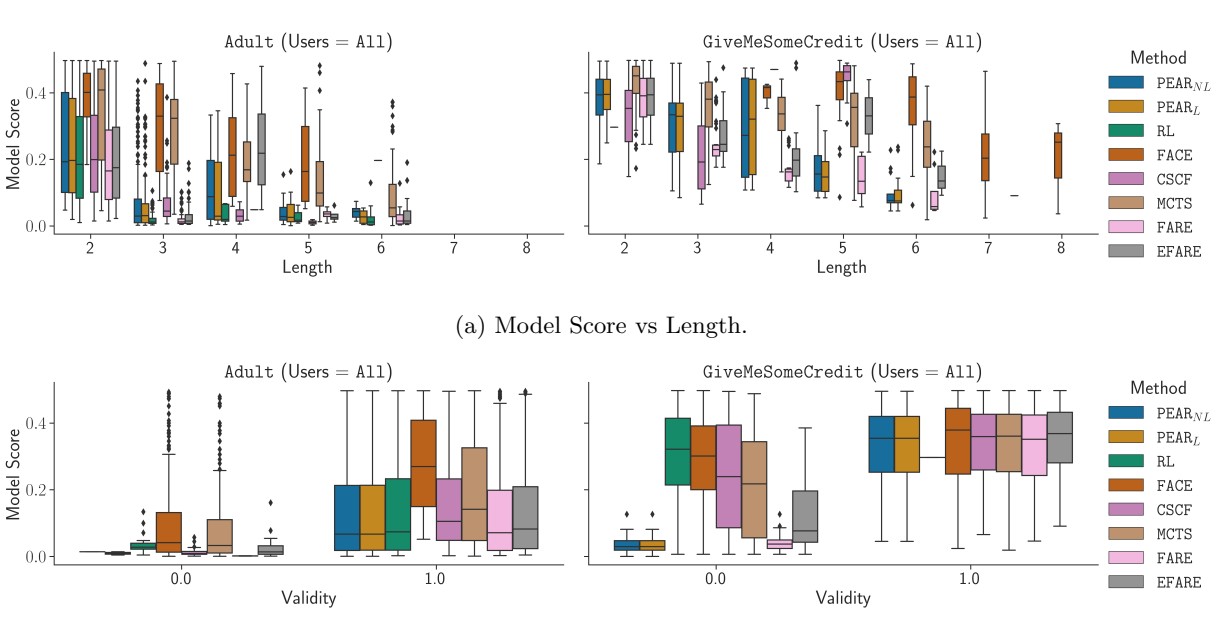

(a) Model Score vs Length.

(b) Model Score vs Validity.

Figure 7: **Model Score vs Length (Top).** In the `Adult` dataset there is a clear correlation between the score and the length of the interventions. Lower scores indicate harder individuals who need longer sequences to obtain recourse. In the `GiveMeSomeCredit` case, we can still see the trend, even if `FACE` and `MCTS` seem to be less susceptible. **Model Score vs Length (Validity) (bottom).** In the `Adult` experiment, all the models have low validity only in the lower-scoring users. In the `GiveMeSomeCredit` experiments, `PEAR` and `FARE` are the only two methods which have low validity only for the `Hard` users, while the others tend to fail more often, irrespectively of the classifier score.

