# OpenReview forum: "Personalized Algorithmic Recourse with Preference Elicitation"
_TMLR — Accepted by TMLR_

### Review · Reviewer_8wdY · 2023-11-13

**Summary Of Contributions:**

This paper studies the problem of "algorithmic recourse" where an algorithm proposes a user a sequence of actions so that this user can improve its classification score. The originality of the algorithm is that the sequence of actions is tailored to some (unknown) user's preference. The authors uses a Bayesian approach to learn this (unknown) preferences by using a weight vector. Some simulations illustrate that this algorithm works;

**Audience:**

Yes

**Broader Impact Concerns:**

This paper does discuss some impact of the paper which seems appropriate.

**Claims And Evidence:**

Yes

**Requested Changes:**

Please correct Weaknesses 1 to 3 above:
1. justify the cost model
2. clarify the role of the numerical section
3. improve notation consistency and general organization of the paper.

**Strengths And Weaknesses:**

While the probleem looks interesting, there are many aspects of the paper that I find unclear.

1. One key aspect of the paper is the cost of an intervention, in Equation (3) and (4). Yet, I do not understand (at all) why should this model make sense. Especially, this cost involves some weight vector"w" -> is it important that the cost has this form (linear in w plus effect of intervention)?  Could the authors explain why the cost is like this? (e.hg. provide references)? Does this make sense for the application that is used as an example?

2. The experiment setting is not clear to me:
- I think that the part about the dataset could serve to illustrate the model and for instance illustrate the concept of dependence graphs, cost of actions... For now, there is very little details about what are "s" and "a" for the considered problems/
- Q1: what is the "rescaling" used? The authors claim that PEAR "minimizes" regret: should the minimum be 0? It probably reaches a "low" regret but does it really "minimize" the regret? Would the "regret" goes to 0 as T goes to infinity?
- Q2: the authors compare their algorithm to other algorithms in the literature. But do these algorithm try to minimize the same cost as the one defined in the paper? If not, it seems quite obvious that they will not perform well for the chosen metrics.

3. Notations and writing should be improved:
- Section 2 "problem statement" is (I think) impossible to understand before having read section 4 because nothing is defined here: what is w? What is s^{(0)}? What is I^{GT}?...
- Section 4 is really confusing, partly because notations are not always consistent (e.g., some "s" become s^{(0)} or s^{(1)} but not always, for instance between (4) and (5)). The non-additivity is quite obvious once the correlation structure is understood. I would encourage the authors to completely rewrite the section between 4 and 4.1 (and probably mix this with section 2), and to try to justify the correlation model. The rest of the section (4.1 to 4.4) is relatively classical but could have its own section.

---

> ### Author Response · Authors · 2023-11-24
> **Response to Reviewer 8wdY [1/2]**
>
> We would like to thank the reviewer for the detailed feedback and suggestions.
>
> **(Q1) One key aspect of the paper is the cost of an intervention, in Equation (3) and (4). Yet, I do not understand (at all) why should this model make sense. Especially, this cost involves some weight vector"w" -> is it important that the cost has this form (linear in w plus effect of intervention)? Could the authors explain why the cost is like this? (e.hg. provide references)? Does this make sense for the application that is used as an example?**
>
> We formalized the cost of an action Equation (3) by following a standard assumption made in decision theory [1] called "generalised additive independence" (GAI) [2] which is shared by many preference elicitation algorithms. It defines utility functions that measure both the contribution of a single feature and the contribution of a subset of features. Following causality theory [3], each subset consists of features causally related to the one we are acting on. In support of this strategy, previous research highlighted how it is impossible to offer optimal recourse recommendations without considering relationships between features [4].
>
> In [4], the authors refer to knowing the impact of an intervention on causally related features (e.g., will getting a new degree increase my salary?). However, it is known that such causal knowledge about the world is hard to obtain unless we make specific restricting assumptions [3]. In our work, rather than considering how features causally change after an intervention, we focus on learning how the "cost" of acting on those features changes after an intervention (e.g., will getting a new degree make it easier to increase my salary?). In our case, the cost of modifying a feature is something we can easily learn by asking the user.
>
> Ultimately, with our setup, we make learning the cost function feasible, and we can also represent complex non-additive behaviour which arises from the sequential nature of recourse. Our formalization in Equation (3) and Equation (4) goes along the lines of previous works [5,6], that address the problem from the same perspective, but assume to know these costs a-priori.
>
> Lastly, we want to stress that PEAR is agnostic to the cost function used. As long as the cost function is parameterized with some learnable parameters, we can use PEAR to learn its parameters in an efficient and sound way. For example, our experiments (Q3, page 9) show that PEAR can perform well even if we consider a misspecified CCS. When the corruption level reaches 100%, PEAR is basically learning a simple weighted linear distance function, where we have no terms coming from the parents. Even in these cases, elicitation still yields an advantage over algorithms that ignore the user’s preferences.
>
> We updated the manuscript to better reflect the rationale behind our formalization.
>
> [1] Keeney, Ralph L., and Howard Raiffa. Decisions with multiple objectives: preferences and value trade-offs. Cambridge University Press, 1993.
>
> [2] Pigozzi, G., Tsoukiàs, A. & Viappiani, P. Preferences in artificial intelligence. Ann Math Artif Intell 77, 361–401 (2016). https://doi.org/10.1007/s10472-015-9475-5
>
> [3] Pearl, Judea. Causality. Cambridge University Press, 2009.
>
> [4] Amir-Hossein Karimi, Julius Von Kügelgen, Bernhard Schölkopf, and Isabel Valera. Algorithmic recourse under imperfect causal knowledge: a probabilistic approach. Advances in Neural Information Processing Systems, 33:265–277, 2020
>
> [5] Philip Naumann and Eirini Ntoutsi. Consequence-aware sequential counterfactual generation. In Joint European Conference on Machine Learning and Knowledge Discovery in Databases, pp. 682–698. Springer, 2021.
>
> [6] Giovanni De Toni, Bruno Lepri, and Andrea Passerini. Synthesizing explainable counterfactual policies for algorithmic recourse with program synthesis. Machine Learning, pp. 1–21, 2023.
>
> **(Q2) The experiment setting is not clear to me: I think that the part about the dataset could serve to illustrate the model and for instance illustrate the concept of dependence graphs, cost of actions... For now, there is very little details about what are "s" and "a" for the considered problems.**
>
> We updated Section 5 to add additional information about the actions and the user state for the experiments. We also extended Appendix A by adding more precise information about the action space used by the PEAR model. Regarding the dependency graph used, we added a missing reference to the Appendix in the main text, where we show the graph used for both experiments (see Figure 4 in Appendix A).

---

> > ### Author Response · Authors · 2023-11-24
> > **Response to Reviewer 8wdY [2/2]**
> >
> > **(Q3) Q1: what is the "rescaling" used? The authors claim that PEAR "minimizes" regret: should the minimum be 0? It probably reaches a "low" regret but does it really "minimize" the regret? Would the "regret" goes to 0 as T goes to infinity?**
> >
> > The rescaling used in Figure 3 is a form of min-max scaling. Since the regret is unbounded, we compute an average of the regrets after t questions by normalizing them taking as a minimum the  $C(I^{(0)} \mid \mathbf{w}^{GT})$, which represents the cost of an intervention computed by taking the population-level estimates of $\mathbf{W}$ and as maximum $C(I^\textrm{GT}\mid \mathbf{w}^{GT})$, which is the cost of the optimal intervention we would have obtained if we had known the ground truth user preferences $\mathbf{w}^\textrm{GT}$.
> >
> > Second, the questions asked by our method are chosen to maximize expected posterior utility, therefore reducing as much as possible regret one step ahead. This is also known as the “myopic value of information”. As more questions are asked, regret will tend to zero. However, a sequential optimization of questions may be more efficient, but it is intractable.
> >
> > **(Q4) Q2: the authors compare their algorithm to other algorithms in the literature. But do these algorithm try to minimize the same cost as the one defined in the paper? If not, it seems quite obvious that they will not perform well for the chosen metrics.**
> >
> > The baselines presented in Section 5 rely on the same cost correlation structure defined in Section 4. For all the competitors, the CCS’s weights are initialized following the same procedure as [7], where they learn population-level costs given some preference data defined by experts, ignoring personal user preferences. The method proposed by [7] is the closest attempt to personalize recourse suggestions by learning a cost function. However, PEAR (our approach) is superior in this since it is the first method which aims to learn personalized user-level cost functions instead, which is something that no other AR method is capable of doing so far. Our experiments show that eliciting the user’s preferences - that is, learning the cost function - is useful. The success of PEAR is due precisely to the fact that it has access to more information than existing approaches, which are not personalized.
> >
> > [7] Kaivalya Rawal and Himabindu Lakkaraju. Beyond individualized recourse: Interpretable and interactive summaries of actionable recourses. Advances in Neural Information Processing Systems, 33:12187–12198, 2020.
> >
> > **(Q5) Notations and writing should be improved:
> > Section 2 "problem statement" is (I think) impossible to understand before having read section 4 because nothing is defined here: what is w? What is s^{(0)}? What is I^{GT}?...
> > Section 4 is really confusing, partly because notations are not always consistent (e.g., some "s" become s^{(0)} or s^{(1)} but not always, for instance between (4) and (5)). The non-additivity is quite obvious once the correlation structure is understood. I would encourage the authors to completely rewrite the section between 4 and 4.1 (and probably mix this with section 2), and to try to justify the correlation model. The rest of the section (4.1 to 4.4) is relatively classical but could have its own section.**
> >
> > We thank the reviewer for the suggestions.
> >
> > Part of the confusion could be due to the fact that $\mathbf{s}^{(i)}$ indicates a state vector at iteration $i$ (with iterations starting from 0), while $s_i$ indicates the $i^{th}$ feature of state vector $\mathbf{s}$ (with features starting from 1).
> > We improved Section 2 to increase its clarity by explaining the missing notations and components of Equation (1) and Equation (2). We also improved Section 4 by expanding the design motivations behind Equation (3) and by improving the notation.

---

### Review · Reviewer_uhhX · 2023-11-14

**Summary Of Contributions:**

The paper proposes a new personalized algorithm for algorithmic recourse that is based on RL principles and an elegant submodular optimization idea. The goal is to find an 'intervention' I, which is a combinatorial set of actions (recommendations) that will be presented to a user so they can work out a way to change a decision initially made by a system (like a credit attribution). Each action of this set has a cost for the user so the system must find a personalized and *least costly* set of action that still manages to have a positive impact on the decision for the user. In general this problem involves solving a complex combinatorial cost minimization problem, but under a specific 'noiseless' response model, the problem becomes submodular and can be solved greedily.

Results are evaluated empirically on several real datasets and show that the algorithm achieves the goal it was trying to reach (minimizing a certain regret with low cost interventions), and outperforms existing state of the art methods.

**Audience:**

Yes

**Broader Impact Concerns:**

This question is addressed in the paper and I have no further concerns.

**Claims And Evidence:**

Yes

**Requested Changes:**

I don't have specific request, but I'd find it interesting if the authors could at least highlight the main interesting theoretical questions that their work could raise and, if they can, explain what are the main technical challenges.

**Strengths And Weaknesses:**

[Disclaimer, I am not an expert in algorithmic recourse and I cannot judge of the contributions compared to the vast literature on the topic]

== Strengths ==
* The paper is impeccably written, which made it accessible to me as a non-expert of the domain
* Experiments are strong and show competitive performance
* The core idea of the algorithm is elegant. It builds on existing work but I would not call it incremental as it introduces several improvement, most importantly the personalized aspect.

== Weaknesses ==

Appart from theoretical principles used to build the algorithm(s), the paper proposes no theoretical guarantees on the defined regret of the algorithm. For instance, I am a bit surprised that the number of users in the data, and perhaps the number of their 'type' does not appear anywhere. Shouldn't it affect how efficient the algorithm is at personalising the interventions?

The influence of T (the iteration budget, or horizon in the online learning parlance) on the quality of the learnt cost and interventions is only studied experimentally, for T=10.

I also wonder how the number of base actions influence these performances.

That being said, I recognise that this may be beyond the scope of the paper.

---

> ### Author Response · Authors · 2023-11-24
> **Response to Reviewer uhhX**
>
> We would like to thank the reviewer for the suggestions and for appreciating the significance of our core message.
>
> **(Q1) Appart from theoretical principles used to build the algorithm(s), the paper proposes no theoretical guarantees on the defined regret of the algorithm. For instance, I am a bit surprised that the number of users in the data, and perhaps the number of their 'type' does not appear anywhere. Shouldn't it affect how efficient the algorithm is at personalising the interventions? The influence of T (the iteration budget, or horizon in the online learning parlance) on the quality of the learnt cost and interventions is only studied experimentally, for T=10. I also wonder how the number of base actions influence these performances.
> I don't have specific request, but I'd find it interesting if the authors could at least highlight the main interesting theoretical questions that their work could raise and, if they can, explain what are the main technical challenges**
>
> Indeed, the development of theoretical insights on the long-term accuracy of cost estimation is an important research direction. We remind the reviewer that our approach (as it is typical in preference elicitation) focuses on the “myopic” value of information (reducing as much as possible regret one step ahead); one main theoretical question currently faced in the preference elicitation domain is the development of methods that compute an optimal elicitation policy (that is, a decision tree), by considering a sequential version of the value of information [1].
>
> We updated the “Benefit and Limitations” section to highlight this fact.
>
> In our experiments, we randomly sampled 300 users from the test sets (for both datasets) and each of them was assigned a weight vector sampled from the population-level prior distribution  $P(\mathbf{W})$. We modelled $P(\mathbf{W})$ as a mixture of Gaussians where each component (which represents one user “type”) has an equal weight. Thus, the user “types” are equally represented in the test set. We updated the manuscript to clarify this point. We agree that the quality of the personalization depends on the user’s type. If the prior used by PEAR fails to cover, for example, less represented individuals, then the personalization might be less accurate. We mentioned this issue in the revised “Broader Impact” section.
>
> [1] Boutilier, Craig. "A POMDP formulation of preference elicitation problems." AAAI/IAAI. 2002.

---

### Review · Reviewer_3FLo · 2023-11-17

**Summary Of Contributions:**

The paper proposes the study of personnalized algorithmic recourse, which is the problem of finding an intervention of minimal cost that overturns an algorithmic decision, where the cost is unknown and depends on the individual who is affected by the decision.

The authors propose PEAR, which is based on a bayesian preference ellicitation algorithm to explore the space of possible actions and costs. The various components of the approach, namely the cost correlation structure which mimics a causal model, the inference of actions based on the current knowledge of the costs and the learning algorithm per se are described. The experiments are made using two standard ML datasets and contain relatively thorough ablations/robustness analyses.

**Audience:**

Yes

**Broader Impact Concerns:**

none that I can think of

**Claims And Evidence:**

Yes

**Requested Changes:**

no requested change. From what I can see and what I understand of TMLR criteria, the submission is up to TMLR publication standards

**Strengths And Weaknesses:**

strength:
* the paper is well-written, I am not up-to-date with the latest literature on AR but the approach seems original to me
* the paper presents a self-contained approach that addresses several technical challenges (learning the parameter of the cost model, exploring action sets and infering best actions) in a meaningful way. The technical novelty is not that large, but the overall approach and makes sense

weaknesses:
* the main weakness I see (as for many if not all papers on AR) is that we are still in the realm of theoretical experiments. It is unclear from the paper how realistic the assumptions and approach is.
* there is no discussion regarding potential biases of the approach (either hidden in the prior, or maybe in the learning algorithm that would favor "majority" costs if they exist). Would there be ways to address/measure those?

---

> ### Author Response · Authors · 2023-11-24
> **Response to Reviewer 3FLo**
>
> We thank the reviewer for the comments and for highlighting the originality of the contribution.
>
> **Q1. the main weakness I see (as for many if not all papers on AR) is that we are still in the realm of theoretical experiments. It is unclear from the paper how realistic the assumptions and approach is. there is no discussion regarding potential biases of the approach (either hidden in the prior, or maybe in the learning algorithm that would favor "majority" costs if they exist). Would there be ways to address/measure those?**
>
> With this paper, we aimed to highlight the necessity of personalizing recourse suggestions, providing a theoretical formulation of the problem and the first interactive approach to learning cost functions from user preferences. As we mentioned in the Conclusions section, the main obstacle in devising realistic experiments is the creation of a scenario where users feel to be unfairly treated. Currently, we also lack the data to build real-world experiments. There is one research work evaluating the usefulness of an interactive AR system from a user perspective, but mostly from a human-computer interaction viewpoint [1]. However, we agree with the reviewer since we also believe that the next steps in AR research would be to focus more on the users, with experiments in the field.
>
> Regarding potential fairness issues, we can see two sources of biases in the case of personalized AR:
> - (1) **Different recourse suggestions when conditioned on a protected attribute.** For example, users who share a similar profile, but differ in some sensitive features (e.g., sex, age, etc.) might obtain different interventions. We believe this might be the most important source of unfairness. In such a scenario, the quality of the learned cost function does not matter, since at a group level, some categories of users will always get costlier interventions. However, this is an issue shared by all methods in the AR literature and is out of the scope of the present paper.
> - (2) **Uninformative prior lacking information about sensitive categories.** The prior used by PEAR might not describe the preferences of less-represented groups, thus making it more difficult to learn such cost functions.
>
> We believe issue (1) relates more to the quality underlying the recourse generator method (e.g., W-FARE, CSCF, etc.). A solution could be employing AR methods that are specifically optimized to satisfy group fairness. However, to the best of our knowledge, this is a little-explored area in the recourse literature. Regarding issue (2), we could imagine focusing our efforts on collecting additional data on sensitive groups to learn more informative priors before running PEAR. Additionally, we could increase the iteration budget (T) to increase the number of pairwise constraints we apply to our MCMC procedure. Nonetheless, as long as users are rational and answer truthfully, the recommendations made by PEAR will be provably cheaper and "fairer" than taking a population-level cost function, as done in the current AR literature.
>
> We updated the manuscript to expand the discussion on fairness and biases in the "Broader Impact" Section.
>
> [1] Wang, Zijie J., et al. "GAM Coach: Towards Interactive and User-centered Algorithmic Recourse." Proceedings of the 2023 CHI Conference on Human Factors in Computing Systems. 2023.

---

### Author Response · Authors · 2023-11-24
**General Comment**

We would like to thank the reviewers for their time and careful reviews of our manuscript.

Following their comments and suggestions, we uploaded a revised version of the manuscript with the changes marked in red colour. In summary:
- Following reviewer #8wdY's suggestions, we improved Section 2 and Section 4 by clarifying the problem statement and the general formal notation. We also expanded the motivations behind our choice of cost function (Equation 3) by drawing more explicit connections to decision theory [1] and causality [2].
- We expanded Appendix A with more explicit information about the action space, the feature space and the DAGs used for the experiments in Section 5.
- Following reviewers #uhhX and #3FLo comments, we expanded the “Broader Impact” (Section 6) to include some possible sources of algorithmic biases hindering the quality of the recourse and some potential mitigation strategies. We also expanded the “Benefits and Limitations” paragraph (4.4) with some additional considerations on the PE’s algorithm used and future directions.

We hope our changes addressed successfully the concerns raised by the reviewers.

[1] Pigozzi, G., Tsoukiàs, A. & Viappiani, P. Preferences in artificial intelligence. Ann Math Artif Intell 77, 361–401 (2016). https://doi.org/10.1007/s10472-015-9475-5

[2] Pearl, Judea. Causality. Cambridge University Press, 2009.

---

### Decision · Action_Editor_ijzk · 2023-12-23

**Recommendation:** Accept as is

**Comment:**

This paper studies algorithmic recourse, the problem of suggesting a sequence of actions to a user impacted by an algorithmic decision to change the decision in the future. The difference from prior works is personalization based on user preferences, which are elicited by interacting with the user. The proposed approach is compared to multiple baselines on two real-world datasets.

The reviewers generally praised the paper for good writing and convincing experiments. The concerns of the reviewers were addressed in the rebuttal and the authors also updated the paper. One limitation of this work is that it does not come with any theoretical guarantees on regret minimization. The reviewers agreed that this is not necessary.

Congratulations for acceptance!

**Audience:**

Yes. The main audience for this paper is fairness in machine learning. This community is large and growing.

**Claims And Evidence:**

Yes. The proposed approach is compared to multiple baselines on two real-world datasets. No theory but the reviewers agreed that this is not necessary.